# Synthesis and Biological Evaluation of Four New Ricinoleic Acid-Derived 1-*O*-alkylglycerols

**DOI:** 10.3390/md18020113

**Published:** 2020-02-15

**Authors:** René Momha, Victor Kuete, Jean-Marie Pagès, Dieudonné Emmanuel Pegnyemb, Paul Mosset

**Affiliations:** 1Univ Rennes, Ecole Nationale Supérieure de Chimie de Rennes, CNRS, ISCR-UMR 6226, F-35000 Rennes, France; 2AGIR, EA 4294, UFR of Pharmacy, Jules Verne University of Picardie, 80037 Amiens, France; 3University of Dschang, Faculty of Science, Department of Biochemistry, P.O. Box 67 Dschang, Cameroon; kuetevictor@yahoo.fr; 4UMR_MD1, U-1261, Aix-Marseille Univ, INSERM, IRBA. Membranes et Cibles Thérapeutiques, Faculté de Pharmacie, 13385 Marseille cedex 05, France; jean-marie.pages@univ-amu.fr; 5Département de ChimieOrganique, Faculté des Sciences, Université de Yaoundé I, BP 812 Yaoundé, Cameroun; pegnyemb@yahoo.com; 6Univ Rennes, CNRS, ISCR (Institut des Sciences Chimiquesde Rennes), UMR 6226, F-35000 Rennes, France

**Keywords:** alkylglycerol (AKG), ricinoleic acid (RA), antimicrobial activity, structure–activity relationship (SAR) studies, antibiotics (gentamicin; tetracycline; ciprofloxacin and ampicillin)

## Abstract

A series of novel substituted 1-*O*-alkylglycerols (AKGs) containing methoxy (**8**), *gem*-difluoro (**9**), azide (**10**) and hydroxy (**11**) group at 12 position in the alkyl chain were synthesized from commercially available ricinoleic acid (**12**). The structures of these new synthesized AKGs were established by NMR experiments as well as from the HRMS and elementary analysis data. The antimicrobial activities of the studied AKGs **8**–**11** were evaluated, respectively, and all compounds exhibited antimicrobial activity to different extents alone and also when combined with some commonly used antibiotics (gentamicin, tetracycline, ciprofloxacin and ampicillin). AKG **11** was viewed as a lead compound for this series as it exhibited significantly higher antimicrobial activity than compounds **8**–**10**.

## 1. Introduction

Natural 1-*O*-alkylglycerols (AKGs) **1** are bioactive ether lipids present in body cells and fluids. They are precursors of ether phospholipids, which participate in structures and functions of membranes in certain cells such as white blood cells or macrophages. AKGs are also found in bone marrow lipids and in milk [1]. Marine sources of AKGs such as the liver oil of certain shark species or rat fish (elasmobranch fishes) contain high levels of these compounds as a mixture of few species varying by length and unsaturation or saturation of the alkyl chain.

The usual composition of alkyl chains in AKGs from Greenland shark (*Centrophorus squamosus*) liver oil (SLO) is as follows: 12:0, 1–2%; 14:0, 1–3%; 16:0, 9–13%; 16:1, n-7, 11–13%; 18:0, 1–5%; 18:1, n-9, 54–68%; 18:1, n-7, 4–6%; and minor species (<1%). Beneficial effects of SLO on health have been recognized in traditional medicine of northern countries involved in fishing such as Japan, Norway and Iceland. In these countries, the ancestral use of SLO was empirical as strengthening or wound healing medication [2].

Experimental studies were performed during the last century, aiming to demonstrate whether AKGs from SLO had biological properties and beneficial effects. Indeed, several studies did observe interesting effects such as hematopoiesis stimulation [3], lowering radiotherapy-induced injuries [4], reducing tumor growth [5] and improving vaccination efficiency [6,7]. However, in most cases, conclusions were mainly impaired by the poor definition of the mixtures used in terms of purity as well as chemical composition. It was then established that the alkyl chain is bound to the glycerol backbone at the *sn-*1 position, thus leading to an *S* configuration at the asymmetric carbon [8] (Figure 1).

To assess the biological activity of each individual AKG, Legrand et al. [9,10,11] reported the antitumor activity (against lung cancer in mice) of each of the six prominent components **2**–**7** of the natural mixture. These derivatives were obtained in pure form by total synthesis and it was observed that the biological activity was heavily dependent upon the unsaturation of the alkyl chain. When this chain was saturated, the corresponding 1-*O*-alkylglycerols **2**–**5** exhibited little or no activity. However, when it was monounsaturated **6**–**7**, a good antitumor activity was observed, thus indicating that the antitumor activity of the natural SLO mixture was heavily related to its unsaturated components (Figure 2).

Currently, resistance to the existing antibiotics and increasing numbers of diseases result in identifying new drug candidates with new forms of activity. Thus, synthesized natural or non-natural AKGs derivatives of natural fatty acids could be one new source of drug delivery systems of antibiotics. Additionally, defined synthetic routes for these targets will facilitate further investigation of biological activities, as natural AKGs were found present in various human cells, but only in trace quantities. Among known natural fatty acids, ricinoleic acid (RA) is one of the major fatty acids occurring in castor oil (almost 90%) [12]. Such a high concentration of this unusual unsaturated fatty acid may be responsible for castor oil’s remarkable healing abilities. It is known to be effective in preventing the growth of numerous species of viruses, bacteria, yeasts and molds [13,14]. Due to the many beneficial effects of this fatty acid component, the use of castor oil can be applied topically to treat a wide variety of health complaints [15], and it also has pharmacological effects on the human gastrointestinal tract [16]. An RA-based glycine derivative was reported to exhibit excellent antimicrobial and anti-biofilm activities against the tested Gram-positive bacterial strains and specifically against various *Candida* strains [17], and it is used for the preparation of several bioactive molecules [18,19,20].

The presence of the hydroxyl group in RA provides a functional group location for performing a variety of chemical reactions including esterification, halogenation, dehydration, alkoxylation and nucleophilic substitution. In this direction, non-natural AKG **8**–**11** derivatives from ricinoleic acid have been synthesized in a stereo-controlled manner. Taking into account RA and natural alkylglycerols’ beneficial effects, herein, we report the synthesis of new non-natural RA-derived methoxy, *gem*-difluorino, azide, and hydroxy-substituted 1-*O*-alkylglycerols **8**–**11** and their respective antimicrobial activities (Figure 3).

## 2. Results and Discussion

### 2.1. Chemistry

Ether lipids bearing a methoxy group at the alkyl chain can be divided into two groups of compounds, namely the methoxylated fatty acids and the methoxy-substituted alkylglycerols [21]. These compounds display interesting biological activities such as antibacterial, antifungal, antitumor and antiviral activities and have been isolated from either bacterial or marine sources, or are mainly of synthetic origin [22]. Hallgren et al. [23] reported that the mixture of methoxylated alkylglycerols made up to 4% of the glyceryl ether content, and in three shark species and three ratfish species they accounted for 0.1% to 0.3% of the total liver lipid content [24].

Methoxy-substituted glyceryl ethers displayed antibacterial and antifungal activities and inhibited some cancer cell lines and metastasis formation in mice [21]. AKGs containing a methoxy group at position 2 in the alkyl chain isolated from the natural SLO mixture of Greenland shark were able to inhibit tumor growth and metastasis formation, and also to stimulate the immunoreactivity in mice [25,26]. Likewise, a synthesized AKG from oleic acid bearing a methoxy group at position 2 in the alkyl chain was reported as an analog of bioactive ether lipid [27].

Therefore, methoxy-substituted AKG **8** was designed as an analog of the AKG **7** with the same alkyl chain length C_18_ and Δ9 unsaturation, but with a methoxy group at position 12 in the alkyl chain to evaluate the beneficial effects of a methoxy group in an other position than 2 in a 1-*O*-alkylglycerol alkyl chain [21,27]. The synthesis of **8** began with the esterification of RA **12** into methyl ricinoleate **13** in 75% yield using boron trifluoride in methanol, along with 4% of a by-product (dimer) **14**, which arises from a subsequent esterification of the secondary alcohol of **13** formed with RA [28]. Compound **14** gave spectroscopic properties (^1^H and ^13^C NMR, as well as correlation spectra) in full agreement with its structure. This was confirmed by HRMS (ESI, *m*/*z*) showing 615.4971 for [M + Na]^+^ (Figure 1). Although it is not strictly necessary, the process could be improved by converting this very minor by-product **14** into **13** by transesterification and thus eliminating it by a subsequent treatment of **13** containing **14** by potassium carbonate in methanol (see experimental section). The secondary alcohol functionality of **13** was then methylated in situ with methyl iodide in the presence of sodium hydroxide in DMSO to provide **15** in 72% yield. This method compares favorably to the hitherto reported cobalt-catalyzed etherification of **13** using diazomethane [29]. Then, **15** was reduced to alcohol **16** in 82% yield using as a reducing agent, Red-Al in Et_2_O at 0 °C. Following this, alcohol **16** underwent mesylation using mesyl chloride in dichloromethane (DCM) with triethylamine, affording **17** in 74% yield, which was in turn alkylated with 2,3-isopropylidene-*sn*-glycerol **18** in the presence of potassium hydroxide and tetra-*n*-butylammonium bromide in DMSO to provide acetonide **19** in 93% yield. Acetonide **19** was hydrolyzed under acidic conditions using a catalytic amount of *p*-toluenesulfonic acid monohydrate in MeOH/H_2_O (9:1) to afford **8** in 99% yield (Figure 1). Appendix A of all synthetic compounds for this sequence is attached in a file as Appendix A.

The hydroxyl functionality of RA makes the castor oil a natural polyol providing oxidative stability to the oil and a relatively long shelf life, compared to other oils, by preventing peroxide formation. As a result, this unique functionality allows the castor oil to be used in industrial applications such as paints, coatings, inks and lubricants [30]. With that in mind, the AKG **11** was designed as an analog of **8**, without a methoxy group at the 12 position in the alkyl chain, to study the influence of the hydroxyl group in the AKG’s alkyl chain, and the structure–activity relationship (SAR) by comparing their biological activity, respectively. The preparation of the AKG **11** with the hydroxyl group required a protection–deprotection sequence to yield the penultimate intermediate alcohol **21**.

Attempt to protect the hydroxy group of **13** using 2-(bromomethyl)naphthalene in the presence of sodium hydroxide and of tetra-*n*-butylammonium bromide in DMSO for 18 h at rt failed. Thereafter, a trace of **20** was observed when chlorotriisopropylsilane in the presence of DIPEA in dichloromethane was used to protect **13** as silyl ether [31]. When DIPEA was replaced by imidazole as a base and in DMF, **20** was obtained in 61% yield. Afterward, **20** was reduced to the penultimate intermediate alcohol **21** in 75% yield using Red-Al in diethyl ether (Figure 2). Compound **21**, upon mesylation conditions using mesyl chloride in DCM, and Et_3_N provided **22** in 76% yield, which was then alkylated under anhydrous conditions with 2,3-isopropylidene-*sn*-glycerol **18** in DMF in the presence of sodium hydride to provide acetonide **23** in 68% yield. Sodium hydride was used as a base instead of KOH to eliminate the by-products formed when reacting **22** with **18**. Following this, silyl ether group was removed using TBAF in THF at rt for 20 h to provide **24** in 92% yield. Easy acetonide cleavage on **24** under acidic conditions (0.05 equiv of *p*-toluenesulfonic acid monohydrate in MeOH/H_2_O (9:1)) gave **11** in 84% yield (Figure 2).

We envisioned that the AKG **9** with a *gem*-difluorinated group in the alkyl chain could exhibit more biological activity than other AKGs studied, as compounds containing a difluoromethylene group were reported to exhibit excellent biological activities [32]. Moreover, introduction of fluorine atoms in molecules heavily modifies their physical, chemical and physiological properties. These fluorinated compounds have found many applications in pharmaceutical and agrochemical fields [33]. Thus, a *gem*-difluorinated **26** key intermediate for the synthesis of **9** was obtained by a classic two-step sequence. Oxidation of **13** by PCC in DCM provided the ketone **25** in 68% yield, which was then subjected to fluorination at rt using (diethylamino)sulfur trifluoride (DAST) in DCM, and the fluorinated product **26** was obtained in 54% yield. Following this, **26** was reduced to alcohol **27** in 79% yield using Red-Al in Et_2_O, which upon treatment with mesyl chloride in DCM in the presence of Et_3_N furnished **28** in 68% yield. Alkylation of **28** in DMSO with 2,3-isopropylidene-*sn*-glycerol (**18**) in the presence of 50% aqueous sodium hydroxide and tetra-*n*-butylammonium bromide gave the expected product **29** in 63% yield, along with 6% of a by-product **30** (Figure 3). Compound **29** was obtained as a pure green oil after column chromatography on florisil gel and showed spectroscopic properties (^1^H and ^13^C NMR as well as correlation spectra) in full agreement with its structure, and confirmation was made by HRMS (ESI, *m*/*z*) that showed 441.3149 for [M + Na]^+^. Sodium hydroxide solution (50%) in H_2_O was used instead of others bases (NaH, KOH) to decrease the formation of the by-product **30**. Acetonide cleavage on compound **29** under acidic conditions using catalytic amount of *p*-toluenesulfonic acid monohydrate in MeOH/H_2_O (9:1) provided **9** in 92% yield (Figure 3).

At last, the AKG **10** was designed as another analog with an azide group at the same position in the alkyl chain to evaluate the beneficial effects of the azide group on the biological activity, and to estimate the dissimilarity between the studied AKGs. Subsequently, **10** was obtained in a classical seven step sequence. Starting under mesylation conditions of methyl ricinoleate **13** using mesyl chloride and Et_3_N in DCM, **31** was obtained in 65% yield, which under substitution of the intermediate mesylate group by S_N_2 substitution reaction using sodium azide in DMSO provided **32** in 80% yield. Dibal in Et_2_O was used to reduce the ester functionality of **32** into alcohol **34**, but surprisingly an aldehyde **33** was formed in 68% yield instead of the expected alcohol **34**. Dibal was chosen as a reducing agent for its non-interaction with the azide group. Moreover, use of another reducing system such as Zn-AlCl_3_ was reported to reduce the azide **32** into the amino derivative [17,19]. Thereafter, aldehyde **33** was then reduced into alcohol **34** in 74% yield using NaBH_4_ in EtOH, which upon reaction with mesyl chloride in DCM in the presence of Et_3_N provided **35** in 59% yield (Figure 4). Following this, alkylation of **35** in DMSO with 2,3-isopropylidene-*sn*-glycerol (**18**) in the presence of 50% aqueous NaOH and tetra-*n*-butylammonium bromide provided a ~1:1 mixture of two compounds: the expected product **36** along with a closer polar by-product **37**, arising from a subsequent elimination of the azide group as indicated in Figure 4.

An attempt to separate the mixture of compounds (**36**,**37**) failed when subjected to maleic anhydride in cyclohexane for 72 h at 45 °C with vigorous stirring. Under these conditions, only **37** was supposed to react with maleic anhydride, leading to a polar product other than **36**, which could ease separation via column chromatography. Compound **36** was then obtained without a trace of by-product **37** by column chromatography on basic alumina gel, and visualization of these two compounds (**36** and **37**) was easily followed on TLC plates as they stained differently with an acidic ethanolic solution of *p*-anisaldehyde. Easy acetonide cleavage on **36** under acidic conditions (*p*-toluenesulfonic acid monohydrate in MeOH/H_2_O (9:1)) provided **10** in 97% yield (Figure 4). Appendix A of all synthetic compounds for this sequence is attached in a file as Appendix A.

The synthesized compounds **8**–**11** were evaluated for their respective antimicrobial activities.

### 2.2. Antimicrobial Activities of AKGs ***8**–**11***

The results of the MIC determination presented in Table 1 indicate detectable values recorded for the (*S*)-3-(((*R,Z*)-12-hydroxyoctadec-9-en-1-yl)oxy)propane-1,2-diol (**11**) on all the eleven studied organisms including bacteria and fungi. All other AKGs, namely (*S*)-3-(((*R*,*Z*)-12-methoxyoctadec-9-en-1-yl)oxy)propane-1,2-diol (**8**), (*S*,*Z*)-3-((12,12-difluorooctadec-9-en-1-yl)oxy)propane-1,2-diol (**9**), and (*S*)-3-(((*S*,*Z*)-12-azidooctadec-9-en-1-yl)oxy)propane-1,2-diol (**10**) showed selective activity. Compound **10** was active on 5 of the 11 (45.5%) whilst AKGs **8** and **9** were active on 8 of the 11 (72.7%) studied microbial species. The lowest MIC value of 19.53 µg/mL was recorded for compound **8** (52.42 µM) on *E. coli* LMP701 and *S. faecalis*, and compound **11** (54.47 µM) on *E. coli* LMP701, *S. typhi* and *C. glabrata*. This lowest MIC value was in some cases equal to that of gentamicin or nystatin on the corresponding microbial species. It appeared from colony count assay (Figure 4) that AKGs **8** and **11** were able to reduce the bacterial concentration after 480 min when tested at the MIC values. A more pronounced effect was reported at 4 × MIC (Figure 5). No growth was observed after treatment with compounds **9** and **10** at 4 × MIC. These data suggest that compounds **8** and **11** might exhibit a killing effect, whilst compounds **9** and **10** could induce a bacteriostatic effect on susceptible microorganisms. Concerning the structure–activity relationship (SAR) studies of AKGs **8**–**11**, it was noticed that the substitution of the hydroxy group at position 12 in the alkyl chain of compound **11** by a methoxy, or a *gem*-difluoro or azide group corresponding to AKGs **8**–**10** respectively, significantly reduced the antimicrobial activity.

AKGs **8**–**11** were also tested in combination with some commonly used antibiotics (Table 2). The results showed that synergistic effects could be obtained in some cases, especially when the AKG **8** was combined with gentamicin, and compound **11** was combined with ciprofloxacin and ampicillin. More than four-fold increase of the activity of these antibiotics was recorded on the three selected bacteria, suggesting that the study should be emphasized on such combinations. The overall activity could be considered as important, mainly when viewed that most of the organisms used were antibiotic resistant. This study therefore provides supportive data for the potential use of the studied 1-*O*-alkylglycerols, in particular AKG **11**, as well as in combination with some antibiotics for the treatment of microbial infections. However, this is to be confirmed by further toxicological studies.

## 3. Materials and Methods

### 3.1. General Experimental Procedures

Moisture sensitive reactions were performed under nitrogen. Anhydrous tetrahydrofuran (THF) and diethyl ether (Et_2_O) were obtained by percolation through a column of a drying resin. Anhydrous *N,N*-dimethylformamide (DMF) over molecular sieves was used as commercially supplied (Acros). Room temperature (rt) means a temperature generally in the range of 18–20 °C. Column chromatography was performed over silica gel Kielselgel 60 (40–60 µm) or basic alumina (Brockmann activity II; basic; pH 10 ± 0.5). Routine monitoring of reactions was carried out using Merck silica gel 60 F_254_ TLC plates (TLC: thin layer chromatography) purchased from Fluka and visualized by UV light (254 nm) inspection followed by staining with an acidic ethanolic solution of *p*-anisaldehyde, or with a solution of phosphomolybdic acid (5 g in 100 mL 95% ethanol). Infrared spectra were recorded with a Thermo Nicolet Avatar 250 FTIR and were reported using the frequency of absorption (cm^−1^). ^1^H NMR spectra (400.13 MHz) and ^13^C NMR spectra (100.61 MHz) were recorded on an Avance 400 Bruker spectrometer using TMS as an internal standard. Multiplicity was tabulated using standard abbreviations: s for singlet, d for doublet, dd for doublet of doublets, t for triplet, q for quadruplet, dtt for doublet of triplets of triplets and m for multiplet (br means broad). Quite obvious first-order ^1^H NMR multiplets were analyzed. As a helpful guidance for this analysis, two articles of Hoye et al. appeared in 1994 and 2002 [34,35]. NMR spectra were processed with zero filling (512 k or 1024 k points). Sometimes resolution enhancement in ^1^H NMR using Traficante facilitated the assignments. Specific rotations were measured on a Perkin Elmer 341 polarimeter, with a cell of 1 dm long and a Na- or Hg-source (Na at 589 nm; Hg at 578 nm, 546 nm, 436 nm and 365 nm), and concentrations were expressed in g/100 mL. High resolution mass spectra (HRMS) were recorded using a MicrO-Tof-Q II spectrometer under electrospray using methanol as solvent. Microanalyses were performed with a CHNS analyzer. The compound 2,3-isopropylidene-*sn*-glycerol (**18**) (≥95% pure) was purchased from Alfa Aesar (France, article # B23037); and ricinoleic acid (**12**) (~80% pure) was purchased from Fluka (Switzerland, article # 83903). Boron trifluoride dimethanol complex (BF_3_^.^2MeOH) was purchased from Acros (Belgium, article # 15890). Products which were used for biological studies were purchased from Maneesh Pharmaceutic PVT (Mumbai, India), Sigma-Aldrich (Johannesburg, South Africa), and Jinling Pharmaceutic Group corp. (Nanjing, China).

### 3.2. Synthesis of Compounds ***8**–**11**,**13**–**17**,**19**–**29*** and ***31**–**36***

#### 3.2.1. (*R*,*Z*)-Methyl 12-hydroxyoctadec-9-enoate (**13**) (Methyl ricinoleate)

To a solution of ricinoleic acid (**12**) (21 g, technical, ~80% pure, ca. 56 mmol) in methanol (140 mL) with stirring, was added BF_3_^.^2MeOH (3.85 mL, 35.5 mmol, 0.63 equiv). Stirring was continued overnight at 50 °C. TLC monitoring showed completion of the reaction mixture after 16 h. Methanol was removed in vacuo, and the resulting oily residue was transferred into a separating funnel with ethyl acetate (100 mL). After washing with brine (3 × 30 mL) and drying over Na_2_SO_4_, ethyl acetate was removed under reduced pressure, and the crude product was purified by column chromatography on silica gel (86 g, 0–4% acetone in petroleum ether) to afford methyl ricinoleate **13** as a colorless oil (16.59 g, 75%) along with a small amount (4%) of a slightly less polar by-product **14**. It eluted after a minor amount of a less polar mixture of methyl oleate and methyl linoleate, R_f_ ca. 0.75 (petroleum ether/acetone 80:20), which was the result of the esterification of the ~15% oleic + linoleic acids, which were contained in technical ricinoleic acid. When potassium carbonate (600 mg) was added to a solution of **13** containing **14** (3.17 g) in methanol (30 mL) with stirring for 42 h (followed by quenching with a solution of citric acid (834 mg) in water (7.5 mL)), the by-product **14** disappeared to afford compound **13** alone, R_f_ = 0.4 (petroleum ether/acetone 80:20).

IR (KBr) ν 3445 (broad, O–H), 3007, 2928, 2855, 1741 (C=O), 1461, 1436, 1246, 1198, 1173, 725 cm^−1^.

^1^H NMR (400 MHz, CDCl_3_): δ 5.56 (dtt, 1H, *J* = 10.9, 7.3, 1.5 Hz, CH=CH–CH_2_–CHOH), 5.40 (dtt, 1H, *J* = 10.9, 7.5, 1.5 Hz, CH=CH–CH_2_–CHOH), 3.67 (s, 3H, CO_2_Me), 3.66–3.56 (m, 1H, CHOH), 2.30 (dd, 2H, *J* = 7.7, 7.4 Hz, CH_2_CO_2_Me), 2.24–2.18 (m, 2H, CHOH–CH_2_–CH=CH), 2.05 (br qd, 2H, *J* = 7.2, 1.2 Hz, CH=CH–CH_2_), 1.62 (br tt, 2H, *J* = 7.5, 7.3 Hz, CH_2_CH_2_CO_2_Me), 1.58 (br d, 1H, *J* = 0.9 Hz, OH, could be s and overlapped at another δ), 1.51–1.41 (m, 3H, CH_2_CHOH and 1H of CH_2_CH_2_CHOH), 1.39–1.23 (m, 15H, 1H of CH_2_CH_2_CHOH, CH_3_(CH_2_)_3_ and (CH_2_)_4_CH_2_CH_2_CO_2_Me), 0.88 (t (approximately t because not first order due to coupling to a rather close CH_2_ at ~1.25 ppm), 3H, *J* = 6.9 Hz, CH_3_).

^13^C NMR (100 MHz, CDCl_3_): δ 174.35 (C_quat_, CO), 133.39 (CH=CH), 125.24 (CH=CH), 71.52 (CHOH), 51.46 (CO_2_CH_3_), 36.87 (CH_2_), 35.37 (CH_2_), 34.10 (CH_2_CO_2_Me), 31.85 (CH_2_CH_2_CH_3_), 29.58 (CH_2_), 29.37 (CH_2_), 29.13 (CH_2_), 29.10 (2 CH_2_), 27.38 (CH_2_), 25.73 (CH_2_), 24.93 (CH_2_CH_2_CO_2_Me), 22.63 (CH_2_CH_3_), 14.10 (CH_3_).

[α]_D_^22^: +4.2; [α]_578_^22^: +4.3; [α]_546_^22^: +4.7; [α]_436_^22^: +7.0; [α]_365_^22^: +8.7 (c 6.00, CHCl_3_),

[α]_D_^22^: +5.7; [α]_578_^22^: +5.9; [α]_546_^22^: +6.6; [α]_436_^22^: +10.7; [α]_365_^22^: +15.4 (c 6.00, acetone).

#### 3.2.2. Physical data for (*R*,*Z*)-(*R*,*Z*)-18-methoxy-18-oxooctadec-9-en-7-yl 12-hydroxyoctadec-9-enoate (**14**)

R_f_ = 0.61 (petroleum ether/acetone 80:20).

IR (KBr) ν 3465 (O–H), 3010, 2928, 2855, 1737 (C=O), 1466, 1456, 1436, 1245, 1194, 1175, 725 cm^−1^.

^1^H NMR (400 MHz, CDCl_3_): δ 5.56 (dtt, 1H, *J* = 10.9, 7.3, 1.5 Hz, CH=CH–CH_2_–CHOH), 5.46 (dtt, 1H, *J* = 10.9, 7.2, 1.5 Hz, CH=CH–CH_2_–CHOCO), 5.40 (dtt, 1H, *J* = 10.9, 7.5, 1.5 Hz, CH=CH–CH_2_–CHOH), 5.32 (dtt, 1H, *J* = 10.9, 7.3, 1.5 Hz, CH=CH–CH_2_–CHOCO), 4.88 (tt, 1H, *J* = 6.3, 6.3 Hz, CHOCO), 3.67 (s, 3H, CO_2_Me), 3.61 (br tt, 1H, *J* = 6.1, 5.7 Hz, CHOH), 2.33–2.24 (m, 6H), 2.23–2.18 (m, 2H), 2.08–1.98 (m, 4H, 2 CH=CH–CH_2_), 1.66–1.43 (m, ~12H, 2 CH_2_CH_2_CO_2_, CH_2_CHOCO, CH_2_CHOH,OH, H_2_O), 1.38–1.21 (m, 32H, 2 CH_3_(CH_2_)_4_ and 2 (CH_2_)_4_CH_2_CH_2_CO_2_), 0.884 (t, 3H, *J* = 6.9 Hz, CH_3_), 0.876 (t, 3H, *J* = 6.9 Hz, CH_3_).

^13^C NMR (100 MHz, CDCl_3_): δ 174.33 (C_quat_, CO_2_Me), 173.59 (CO_2_CH), 133.39 (CH=CH), 132.53 (CH=CH), 125.22 (CH=CH), 124.35 (CH=CH), 73.70 (CHOCO), 71.51 (CHOH), 51.46 (CO_2_CH_3_), 36.87 (CH_2_), 35.38 (CH_2_), 34.68 (CH_2_), 34.10 (CH_2_CO_2_Me), 33.65 (CH_2_CO_2_CH), 32.00 (CH_2_), 31.85 (CH_2_CH_2_CH_3_), 31.76 (CH_2_), 29.62 (CH_2_), 29.53 (CH_2_), 29.36 (CH_2_), 29.19 (CH_2_), 29.18 (CH_2_), 29.16 (CH_2_), 29.14 (CH_2_), 29.13 (CH_2_), 29.12 (2 CH_2_), 27.40 (CH_2_), 27.34 (CH_2_), 25.73 (CH_2_), 25.37 (CH_2_), 25.10 (CH_2_), 24.95 (CH_2_CH_2_CO_2_Me), 22.63 (CH_2_), 22.59 (CH_2_), 14.10 (CH_3_), 14.08 (CH_3_).

[α]_D_^21^: +17.4; [α]_578_^21^: +18.1; [α]_546_^21^: +20.7; [α]_436_^21^: +34.9; [α]_365_^21^: +54.8 (c 2.56, CHCl_3_).

HRMS (ESI, *m*/*z*) calculated for C_37_H_68_O_5_Na [M + Na]^+^: 615.4964, found: 615.4971.

#### 3.2.3. (*R*,*Z*)-Methyl 12-methoxyoctadec-9-enoate (**15**)

In a flask containing a solution of **13** (625 mg, 2.0 mmol), tetra-*n*-butylammonium bromide (709.2 mg, 2.2 mmol, 1.2 equiv) in DMSO (2.0 mL) with stirring, was added finely crushed (with a mortar and pestle) sodium hydroxide (250 mg, 6 mmol, 3 equiv) and methyl iodide (0.63 mL, 10 mmol, 5 equiv). The reaction flask was flushed under nitrogen, tightly stoppered and protected from light by wrapping with an aluminum foil. After stirring overnight for 18 h, TLC monitoring showed that the reaction was mostly done. An aqueous solution of 10% citric acid (10 mL) was added, and extraction was done with petroleum ether/EtOAc (80:20). Organic layers were dried over Na_2_SO_4_, and solvent was removed under reduced pressure. Then, the residue was purified by column chromatography on basic alumina (5 g, 0%–0.2% acetone in petroleum ether) to afford **15** as a colorless oil (471 mg, 72%). *R*_f_ = 0.6 (petroleum ether/acetone 90:10).

IR (KBr) ν 3465 (small, harmonic of C=O), 3009, 2929, 2855, 1742 (C=O), 1463, 1456, 1436, 1360, 1245, 1195, 1173, 1099 (C–O of OMe), 725 cm^−1^.

^1^H NMR (400 MHz, CDCl_3_): δ 5.50–5.34 (m, 2H: CH=CH partly distorted due to strong coupling at 5.45 and 5.38 ppm (dtt, *J* = 10.9, 6.9, 1.4 Hz)), 3.67 (s, 3H, CO_2_Me), 3.34 (s, 3H, CHOCH_3_), 3.17 (tt, 1H, *J* = 6.2, 5.5 Hz, CHOMe), 2.30 (dd, 2H, *J* = 7.7, 7.4 Hz, CH_2_CO_2_Me), 2.30–2.17 (m, 2H, CHOMe–CH_2_–CH=CH), 2.03 (br q, 2H, *J* = 6.7 Hz, CH=CH–CH_2_), 1.67–1.57 (m, 2H, CH_2_CH_2_CO_2_Me), 1.49–1.41 (m, 2H, CH_2_CHOMe), 1.40–1.21 (m, 16H, CH_3_(CH_2_)_4_ and (CH_2_)_4_CH_2_CH_2_CO_2_Me), 0.88 (t, 3H, *J* = 6.9 Hz, CH_3_).

^13^C NMR (100 MHz, CDCl_3_): δ 174.33 (C_quat_, CO), 131.73 (CH=CH), 125.42 (CH=CH), 80.99 (CHOMe), 56.58 (CHOCH_3_), 51.46 (CO_2_CH_3_), 34.11 (CH_2_CO_2_Me), 33.57 (CH_2_), 31.88 (CH_2_CH_2_CH_3_), 31.05 (CH_2_), 29.56 (CH_2_), 29.50 (CH_2_), 29.18 (CH_2_), 29.15 (CH_2_), 29.13 (CH_2_), 27.41 (CH_2_), 25.36 (CH_2_), 24.95 (CH_2_CH_2_CO_2_Me), 22.65 (CH_2_CH_3_), 14.10 (CH_3_).

[α]_D_^17.5^: +13.6; [α]_578_^17.5^: +14.1; [α]_546_^17.5^: +16.1; [α]_436_^17.5^: +27.5; [α]_365_^17.5^: +43.2 (c 5.00, CHCl_3_)

[α]_D_^17.5^: +16.2; [α]_578_^17.5^: +16.9; [α]_546_^17.5^: +19.2; [α]_436_^17.5^: +32.6 (neat liquid).

#### 3.2.4. (*R*,*Z*)-12-Methoxyoctadec-9-en-1-ol (**16**)

Red-Al (0.52 mL, ~3 M in toluene, 1.56 mmol, 1.2 equiv) was added dropwise to a cooled solution of **15** (422 mg, 1.29 mmol) in anhydrous Et_2_O (4 mL) at 0 °C with stirring and under nitrogen. After the addition, the stirring was continued overnight for 18 h at 0 °C (use of a Dewar with ice-cooling). TLC monitoring confirmed disappearance of the starting material. A solution of citric acid (400 mg) in distilled water (5 mL) was added to the reaction mixture, which was allowed to stir again for 30 min. Extraction was done with petroleum ether/EtOAc (80:20). Organic layers were dried over Na_2_SO_4_, and solvent was evaporated under reduced pressure. The crude product was then purified by column chromatography on basic alumina (5 g, 0%–3% acetone in petroleum ether) to afford **16** as a colorless oil (318 mg, 82%). R_f_ = 0.32 (petroleum ether/acetone 85:15).

IR (KBr) ν 3372 (broad, O–H), 3009, 2927, 2855, 1464, 1456, 1377, 1357, 1099 (C–O of OMe), 1058, 724 cm^−1^.

^1^H NMR (400 MHz, CDCl_3_): δ 5.50–5.34 (m, 2H: CH=CH partly distorted due to strong coupling at 5.46 and 5.38 ppm (dtt, *J* = 10.9, 7.0, 1.4 Hz)), 3.64 (t, 2H, *J* = 6.6 Hz, CH_2_OH), 3.34 (s, 3H, CHOCH_3_), 3.17 (tt, 1H, *J* = 6.2, 5.5 Hz, CHOMe), 2.31–2.17 (m, 2H, CHOMe–CH_2_–CH=CH), 2.03 (br q, 2H, *J* = 6.7 Hz, CH=CH–CH_2_), 1.66 (br s, 1H, OH), 1.56 (br tt, 2H, *J* = 7.5, 6.6 Hz, CH_2_CH_2_OH), 1.49–1.41 (m, 2H, CH_2_CHOMe), 1.41–1.23 (m, 18H, CH_3_(CH_2_)_4_ and (CH_2_)_5_CH_2_CH_2_OH), 0.88 (t, 3H, *J* = 6.9 Hz, CH_3_).

^13^C NMR (100 MHz, CDCl_3_): δ 131.81 (CH=CH), 125.38 (CH=CH), 81.02 (CHOMe), 63.06 (CH_2_OH), 56.58 (CHOCH_3_), 33.57 (CH_2_), 32.80 (CH_2_CH_2_OH), 31.88 (CH_2_), 31.07 (CH_2_), 29.61 (CH_2_), 29.50 (2 CH_2_), 29.40 (CH_2_), 29.27 (CH_2_), 27.43 (CH_2_), 25.74 (CH_2_), 25.36 (CH_2_), 22.65 (CH_2_), 14.12 (CH_3_).

[α]_D_^23.5^: +12.8; [α]_578_^23.5^: +13.4; [α]_546_^23.5^: +15.2; [α]_436_^23.5^: +25.9; [α]_365_^23.5^: +40,6 (c 5.02, CHCl_3_).

[α]_D_^23.5^: +17.0; [α]_578_^23.5^: +17.4; [α]_546_^23.5^: +19.8; [α]_436_^23.5^: +33.6; [α]_365_^23.5^: +53.2 (c 5.02, acetone).

[α]_D_^23^: +15.7; [α]_578_^23^: +16.4; [α]_546_^23^: +18.6; [α]_436_^23^: +31.7 (neat liquid).

HRMS (ESI, *m*/*z*) calculated for C_19_H_38_O_2_Na [M + Na]^+^: 321.4892, found: 321.4886.

#### 3.2.5. (*R*,*Z*)-12-Methoxyoctadec-9-en-1-yl methanesulfonate (**17**)

To a stirred solution of **16** (6.64 g, 22.2 mmol), Et_3_N (4.7 mL, 33.4 mmol, 1.5 equiv) in DCM (67 mL) under nitrogen at −30 °C, mesyl chloride (2.2 mL, 28 mmol, 1.25 equiv) in DCM (9 mL) was added dropwise. The addition of mesyl chloride was completed by rinsing with DCM (3 × 0.3 mL). The corresponding mixture was then stirred for 15 h at −30 °C. TLC monitoring (elution with DCM, mesylate showed far bigger R_f_ than starting alcohol with this eluent) showed completion of the reaction, and distilled water (75 mL) was added to quench the reaction. Extraction was done with DCM. Organic layers were washed with brine and dried over Na_2_SO_4_. Solvent was removed under reduced pressure to provide the crude product as a light yellow oil. The crude product was then purified by column chromatography on silica gel (10 g, 0%–2% acetone in petroleum ether) to provide **17** as a colorless oil (5.40 g, 74%). R_f_ = 0.45 (petroleum ether/acetone 80:20).

IR (KBr) ν 3011, 2928, 2855, 1464, 1358, 1177 (S=O), 1098 (C–O of OMe), 974, 945, 831, 724, 529 cm^−1^.

^1^H NMR (400 MHz, CDCl_3_): δ 5.50–5.34 (m, 2H: CH=CH partly distorted due to strong coupling at 5.46 and 5.38 ppm (dtt, *J* = 10.9, 6.9, 1.3 Hz)), 4.22 (t, 2H, *J* = 6.6 Hz, CH_2_OMs), 3.34 (s, 3H, CHOCH_3_), 3.17 (tt, 1H, *J* = 6.1, 5.5 Hz, CHOMe), 3.00 (s, 3H, SO_2_CH_3_), 2.31–2.17 (m, 2H, CHOMe–CH_2_–CH=CH), 2.03 (br q, 2H, *J* = 6.7 Hz, CH=CH–CH_2_), 1.75 (br tt, 2H, *J* = 7.5, 6.6 Hz, CH_2_CH_2_OMs), 1.50–1.22 (m, 20H, CH_3_(CH_2_)_5_ and (CH_2_)_5_CH_2_CH_2_OMs), 0.88 (t, 3H, *J* = 6.9 Hz, CH_3_).

^13^C NMR (100 MHz, CDCl_3_): δ 131.70 (CH=CH), 125.47 (CH=CH), 80.98 (CHOMe), 70.16 (CH_2_OMs), 56.59 (CHOCH_3_), 37.38 (SO_2_CH_3_), 33.57 (CH_2_), 31.88 (CH_2_), 31.08 (CH_2_), 29.57 (CH_2_), 29.49 (CH_2_), 29.34 (CH_2_), 29.21 (CH_2_), 29.13 (CH_2_), 29.02 (CH_2_), 27.41 (CH_2_), 25.43 (CH_2_), 25.36 (CH_2_), 22.64 (CH_2_), 14.10 (CH_3_).

[α]_D_^23.5^: +13.7; [α]_578_^23.5^: +14.1; [α]_546_^23.5^: +16.0; [α]_436_^23.5^: +27.5; [α]_365_^23.5^: +43.4 (c 5.01, acetone).

[α]_D_^23.5^: +10.3; [α]_578_^23.5^: +10.5; [α]_546_^23.5^: +11.9; [α]_436_^23.5^: +20.3; [α]_365_^23.5^: +31.5 (c 2.60, CHCl_3_).

[α]_D_^23.5^: +13.7; [α]_578_^23.5^: +14.3; [α]_546_^23.5^: +16.3; [α]_436_^23.5^: +27.2 (neat liquid).

#### 3.2.6. (*R*)-4-((((*R*,*Z*)-12-Methoxyoctadec-9-en-1-yl)oxy)methyl)-2,2-dimethyl-1,3-dioxolane (**19**)

In a flask containing a solution of **17** (4.97 g, 13.2 mmol), *n*-Bu_4_NBr (1.06 g, 3.3 mmol, 0.25 equiv), and finely crushed (with a mortar and pestle) potassium hydroxide (3.48 g, 52.75 mmol, ~85% KOH, 4 equiv) in DMSO (26.4 mL) with stirring and under nitrogen at rt for 10 min, was added (*R*)-solketal (**18**) (2.07 g, ≥95% pure, 14.9 mmol, 1.13 equiv), and the corresponding mixture was stirred overnight for 14 h at 35 °C. TLC monitoring showed completion of the reaction and distilled water (50 mL) was added to the reaction mixture. Extraction was done with petroleum ether/EtOAc (80:20). Organic layers were washed with brine and dried over Na_2_SO_4_. Solvent was removed under reduced pressure. The crude product was then purified by column chromatography on basic alumina (25 g, 0%–1% acetone in petroleum ether) to afford **19** as a colorless oil (4.65 g, 93%). R_f_ = 0.45 (petroleum ether/acetone 95:5).

IR (KBr) ν 2985, 2929, 2856, 2821, 1466, 1456, 1379, 1369, 1256, 1237, 1214, 1118, 1100 (C–O of OMe), 1056, 847, 724, 514 cm^−1^.

^1^H NMR (400 MHz, CDCl_3_): δ 5.50–5.34 (m, 2H: CH=CH partly distorted due to strong coupling at 5.46 and 5.38 ppm (dtt, *J* = 10.9, 6.9, 1.3 Hz)), 4.27 (dddd (apparent tt), 1H, *J* = 6.4, 6.4, 5.7, 5.6 Hz, CH–O in dioxolane), 4.06 (dd, 1H, *J* = 8.2, 6.4 Hz, CH_2_O in dioxolane), 3.73 (dd, 1H, *J* = 8.2, 6.4 Hz, CH_2_O in dioxolane), 3.54–3.39 (m, 4H, CH_2_OCH_2_O with 1H dd at 3.52 ppm, *J* = 9.9, 5.7 Hz and 1H dd at 3.42 ppm, *J* = 9.9, 5.6 Hz), 3.34 (s, 3H, CHOCH_3_), 3.17 (br tt, 1H, *J* = 6.2, 5.4 Hz, CHOMe), 2.31–2.17 (m, 2H, CHOMe–CH_2_–CH=CH), 2.03 (br q, 2H, *J* = 6.7 Hz, CH=CH–CH_2_), 1.62–1.52 (m, 2H, CH_2_CH_2_O), 1.50–1.41 (m, 2H, CH_2_CHOMe), 1.42 (q, 3H, *J* = 0.7 Hz (w coupling), CH_3_), 1.36 (q, 3H, *J* = 0.7 Hz (w coupling), CH_3_), 1.39–1.23 (m, 18H, CH_3_(CH_2_)_4_ and (CH_2_)_3_CH_2_CH_2_O), 0.88 (t, 3H, *J* = 6.9 Hz, CH_3_).

^13^C NMR (100 MHz, CDCl_3_): δ 131.82 (CH=CH), 125.35 (CH=CH), 109.37 (CMe_2_), 80.99 (CHOMe), 74.76 (CH–O), 71.88 and 71.83 (CH_2_OCH_2_(CH_2_)_7_), 66.93 (CH_2_OCMe_2_), 56.58 (CHOCH_3_), 33.57 (CH_2_), 31.88 (CH_2_), 31.04 (CH_2_), 29.63 (CH_2_), 29.56 (CH_2_), 29.51 (CH_2_), 29.50 (CH_2_), 29.45 (CH_2_), 29.30 (CH_2_), 27.44 (CH_2_), 26.78 (C–CH_3_), 26.06 (CH_2_), 25.43 (C–CH_3_), 25.36 (CH_2_), 22.64 (CH_2_), 14.11 (CH_3_).

[α]_D_^18^: +4.4; [α]_578_^18^: +4.1; [α]_546_^18^: +4.7; [α]_436_^18^: +7.3; [α]_365_^18^: +10.7 (c 3.01, acetone).

[α]_D_^18^: +2.7; [α]_578_^18^: +2.8; [α]_546_^18^: +3.0; [α]_436_^18^: +4.8; [α]_365_^18^: +6.3 (c 2.42, CHCl_3_).

HRMS (ESI, *m*/*z*) calculated for C_25_H_48_O_4_Na [M + Na]^+^: 435.3450, found: 435.3452.

Elementary analysis calculated for C_25_H_48_O_4_: C, 72.77; H, 11.72, found: C, 73.23; H, 11.94.

#### 3.2.7. (*S*)-3-(((*R*,*Z*)-12-Methoxyoctadec-9-en-1-yl)oxy)propane-1,2-diol (**8**)

To a solution of acetonide **19** (4.65 g, 11.2 mmol) in methanol (45.1 mL), was added *p*-toluenesulfonic acid monohydrate (107.3 mg, 0.55 mmol, 0.05 equiv) and distilled water (4.5 mL). The flask was then purged with nitrogen, stoppered and dipped in a preheated bath at 60 °C. Stirring was maintained for 5 h at 60 °C. TLC monitoring showed completion of the reaction, and sodium bicarbonate (52.1 mg, 0.62 mmol, 0.055 equiv) was added. Stirring was continued for 1 h at 60 °C. Methanol was then removed under reduced pressure, and the crude product was purified by column chromatography on silica gel (20 g, 0%–5% acetone in petroleum ether and then petroleum ether + 5% acetone + 12% methanol) to afford **8** as a colorless oil (4.16 g, 99%). R_f_ = 0.03 (petroleum ether/acetone 90:10).

^1^H NMR (400 MHz, CDCl_3_): δ 5.50–5.34 (m, 2H: CH=CH partly distorted due to strong coupling at 5.46 and 5.38 ppm (dtt, *J* = 10.9, 7.0, 1.4 Hz)), 3.90–3.83 (m, 1H, CHOH), 3.72 (broad ddd, 1H, *J* = 11.3, 6.7 (with OH), 3.9 Hz, CH_2_OH), 3.65 (broad ddd, 1H, *J* = 11.3, 5.1 (with OH), 4.9 Hz, CH_2_OH), 3.56–3.42 (m, 4H, CH_2_OCH_2_ with 1H dd at 3.54 ppm, *J* = 9.7, 4.0 Hz), 3.34 (s, 3H, CHOCH_3_), 3.17 (tt, 1H, *J* = 6.2, 5.4 Hz, CHOMe), 2.67 (d, 1H, *J* = 5.0 Hz, CHOH), 2.28 (broad dd, 1H, *J* = 6.7, 5.1 Hz, resolution ω = 1.3 Hz, CH_2_OH), 2.30–2.17 (m, 2H, CHOMe–CH_2_–CH=CH), 2.03 (br q, 2H, *J* = 6.7 Hz, CH=CH–CH_2_), 1.57 (broad tt, 2H, *J* = 7.1, 6.7 Hz, CH_2_CH_2_O), 1.50–1.41 (m, 2H, CH_2_CHOMe), 1.41–1.23 (m, 18H, CH_3_(CH_2_)_4_ and (CH_2_)_5_CH_2_CH_2_O), 0.88 (t, 3H, *J* = 6.9 Hz, CH_3_).

^13^C NMR (100 MHz, CDCl_3_): δ 131.81 (CH=CH), 125.36 (CH=CH), 81.00 (CHOMe), 72.51 and 71.82 (CH_2_OCH_2_(CH_2_)_7_), 70.41 (CHOH), 64.29 (CH_2_OH), 56.59 (CHOCH_3_), 33.54 (CH_2_), 31.88 (CH_2_), 31.04 (CH_2_), 29.60 (CH_2_), 29.57 (CH_2_), 29.49 (CH_2_), 29.46 (CH_2_), 29.40 (CH_2_), 29.26 (CH_2_), 27.42 (CH_2_), 26.06 (CH_2_), 25.35 (CH_2_), 22.64 (CH_2_), 14.11 (CH_3_).

[α]_D_^22.5^: +9.8; [α]_578_^22.5^: +9.6; [α]_546_^22.5^: +11.1; [α]_436_^22.5^: +19.6; [α]_365_^22.5^: +31.5 (c 1.10, acetone).

HRMS (ESI, m/z) calculated for C_22_H_44_O_4_Na [M + Na]^+^: 395.3137, found: 395.3132.

#### 3.2.8. (*R*,*Z*)-Methyl 12-((triisopropylsilyl)oxy)octadec-9-enoate (**20**)

To a vigorously stirred solution of **13** (625 mg, 2.0 mmol) and imidazole (334 mg, 4.9 mmol, 2.45 equiv) in DMF (1.6 mL), which was cooled under nitrogen at 0 °C, was added dropwise triisopropylsilyl chloride (0.53 mL, 97% pure, 2.4 mmol, 1.2 equiv). The corresponding mixture was allowed to stir for 48 h at rt. TLC monitoring showed completion of the reaction. Petroleum ether/EtOAc (80:20) was added. After washing with brine and drying over MgSO_4_, solvent was removed under reduced pressure, and the residue was purified by column chromatography on silica gel (5 g, 0%–0.2% acetone in petroleum ether) to provide **20** as a colorless oil (574.7 mg, 61%). *R*_f_ = 0.73 (petroleum ether/acetone 95:5).

IR (KBr) ν 3006, 2929, 2865, 1744, 1464, 1436, 1366, 1096, 883, 678 cm^−1^.

^1^H NMR (400 MHz, CDCl_3_): δ 5.46–5.36 (m, 2H), 3.83 (tt, 1H, *J* = 5.8, 5.6 Hz), 3.67 (s, 3H), 2.30 (dd, 2H, *J* = 7.8, 7.4 Hz), 2.27–2.22 (m, 2H), 2.06–1.97 (m, 2H), 1.67–1.57 (m, 2H), 1.54–1.38 (m, 2H), 1.38–1.20 (m, 16H), 1.06 (s, 21H, Si(CH(CH_3_)_2_)_3_, due to the shielding effect on neighboring CH, CH and CH_3_ of isopropyl groups being very close, so coupling was not seen and these signals were superimposed, HSQC showed CH at 1.061 ppm and CH_3_ at 1.059 ppm), 0.88 (t, 3H, *J* = 6.9 Hz, CH_3_).

^13^C NMR (100 MHz, CDCl_3_): δ 174.33 (CO), 131.31 (CH=CH), 125.72 (CH=CH), 72.26 (CH), 51.46 (CH_3_), 36.56 (CH_2_), 34.68 (CH_2_), 34.11 (CH_2_), 31.92 (CH_2_), 29.64 (2 CH_2_), 29.20 (CH_2_), 29.19 (CH_2_), 29.15 (CH_2_), 27.51 (CH_2_), 24.96 (CH_2_), 24.81 (CH_2_), 22.64 (CH_2_), 18.21 (6 CH_3_), 14.11 (CH_3_), 12.63 (3 CH).

[α]_D_^18^: +12.3; [α]_578_^18^: +12.5; [α]_546_^18^: +14.2; [α]_436_^18^: +24.3; [α]_365_^18^: +38.6 (c 4.00, acetone).

[α]_D_^18^: +10.6; [α]_578_^18^: +11.0; [α]_546_^18^: +12.6; [α]_436_^18^: +21.5; [α]_365_^18^: +33.7 (c 4.00, CHCl_3_).

HRMS (ESI, *m*/*z*) calculated for C_26_H_51_O_3_Si [M–C_2_H_5_]^+^: 439.3607, found: 439.3607, calculated for C_25_H_49_O_3_Si [M–*i*Pr]^+^: 425.3451, found: 425.3455, calculated for C_24_H_45_O_2_Si [M–*i*Pr-MeOH]^+^: 393.3189, found: 393.3177.

#### 3.2.9. (*R*,*Z*)-12-((Triisopropylsilyl)oxy)octadec-9-en-1-ol (**21**)

In a flame-dried two-necked flask, a solution of **20** (574.7 mg, 1.22 mmol) in anhydrous Et_2_O (3.7 mL) was cooled at 0 °C under nitrogen. A solution of Red-Al (65% in toluene, ~3 M, 0.5 mL, 1.5 mmol, 1.23 equiv) was added dropwise under stirring. Stirring was continued for 5 h at 0 °C. TLC monitoring confirmed completion of the reaction. Citric acid (400 mg) and distilled water (5 mL) were added to the mixture and stirring was continued for 30 min. Extraction was then done with petroleum ether/EtOAc (80:20), and organic layers were dried over Na_2_SO_4_. Solvent was evaporated under reduced pressure, and the crude product was purified by column chromatography on silica gel (2.5 g, 0%–0.5% acetone in petroleum ether) to afford **21** as a colorless oil (509.6 mg, 94%). *R*_f_ = 0.39 (petroleum ether/acetone 85:15).

^1^H NMR (400 MHz, CDCl_3_): δ 5.47–5.36 (m, 2H), 3.83 (tt, 1H, *J* = 6.0, 5.6 Hz), 3.64 (t, 2H, *J* = 6.6 Hz, CH_2_OH), 2.27–2.22 (m, 2H), 2.02 (br q, 2H, *J* = 6.5 Hz), 1.61–1.52 (m, 3H, CH_2_ and OH), 1.52–1.40 (m, 2H), 1.40–1.19 (m, 18H), 1.06 (s, 21H), 0.88 (t, 3H, *J* = 6.9 Hz, CH_3_).

^13^C NMR (100 MHz, CDCl_3_): δ 131.38 (CH=CH), 125.68 (CH=CH), 72.27 (CH), 63.10 (CH_2_), 36.55 (CH_2_), 34.68 (CH_2_), 32.81 (CH_2_), 31.92 (CH_2_), 29.69 (CH_2_), 29.64 (CH_2_), 29.54 (CH_2_), 29.42 (CH_2_), 29.32 (CH_2_), 27.54 (CH_2_), 25.74 (CH_2_), 24.81 (CH_2_), 22.64 (CH_2_), 18.21 (6 CH_3_), 14.11 (CH_3_), 12.63 (3 CH).

#### 3.2.10. (*R*,*Z*)-12-((Triisopropylsilyl)oxy)octadec-9-en-1-yl methanesulfonate (**22**)

To a solution of **21** (509.6 mg, 1.15 mmol) and Et_3_N (0.245 mL, 1.75 mmol, 1.5 equiv) in DCM (4.6 mL) with stirring and under nitrogen cooled at −50 °C, was added dropwise mesyl chloride (0.112 mL, 1.45 mmol, 1.25 equiv) in DCM (0.6 mL). Transfer of mesyl chloride was completed by rinsing twice with a few drops of DCM. The reaction mixture was then allowed to warm up slowly in 2 h up to −5 °C. TLC monitoring (elution with DCM) showed completion of the reaction, and distilled water (5.8 mL) was added to quench the reaction. Extraction was done with DCM. Organic layers were washed with brine and dried over Na_2_SO_4_. DCM was removed under reduced pressure, and the crude product was purified by column chromatography on silica gel (4 g, 0%–1% acetone in petroleum ether) to afford **22** as a colorless oil (453 mg, 76%). R_f_ = 0.41 (petroleum ether/acetone 80:20).

^1^H NMR (400 MHz, CDCl_3_): δ 5.47–5.37 (m, 2H), 4.22 (t, 2H, *J* = 6.6 Hz, CH_2_OMs), 3.83 (tt, 1H, *J* = 6.0, 5.6 Hz), 3.00 (s, 3H), 2.29–2.19 (m, 2H), 2.08–1.96 (m, 2H), 1.75 (dq, 2H, *J* = 8.2, 6.6 Hz), 1.54–1.20 (m, 20H), 1.06 (s, 21H), 0.88 (t, 3H, *J* = 6.9 Hz, CH_3_).

^13^C NMR (100 MHz, CDCl_3_): δ 131.28 (CH=CH), 125.75 (CH=CH), 72.25 (CH), 70.16 (CH_2_), 37.37 (CH_3_), 36.56 (CH_2_), 34.69 (CH_2_), 31.91 (CH_2_), 29.65 (CH_2_), 29.63 (CH_2_), 29.37 (CH_2_), 29.25 (CH_2_), 29.14 (CH_2_), 29.04 (CH_2_), 27.51 (CH_2_), 25.44 (CH_2_), 24.81 (CH_2_), 22.64 (CH_2_), 18.22 (6 CH_3_), 14.11 (CH_3_), 12.64 (3 CH).

HRMS (ESI, m/z) calculated for C_28_H_58_O_4_NaSiS [M + Na]^+^: 541.3723, found: 541.3724.

#### 3.2.11. (((*R*,*Z*)-18-(((*R*)-2,2-Dimethyl-1,3-dioxolan-4-yl)methoxy)octadec-9-en-7-yl)oxy)triisopropylsilane (**23**)

A 60% dispersion of sodium hydride in mineral oil (29.5 mg, 0.7 mmol, 2.5 equiv) was washed three times with petroleum ether under nitrogen and with stirring. Anhydrous DMF (0.2 mL) was then added and the mixture was cooled at 0 °C. Following this, a solution of 2,3 isopropylidene-*sn*-glycerol **18** (50.2 mg, ≥95% pure, 0.36 mmol, 1.25 equiv) in DMF (0.2 mL) was added dropwise to the mixture, and the flask containing **18** was rinsed with DMF (2 × 0.1 mL). The corresponding mixture was allowed to stir for 10 min at 0 °C, and a solution of **22** (153 mg, 0.29 mmol) in DMF (0.2 mL) was added to the resulting white suspension. Transfer of **22** was completed by rinsing with DMF (2 × 0.1 mL). The mixture was then vigorously stirred overnight for 16 h at rt. TLC monitoring showed completion of the reaction, and 10% aqueous ammonium acetate was added as a buffer. Extraction was done with petroleum ether. Organic layers were dried over Na_2_SO_4_. Solvent was removed under reduced pressure, and the crude product was purified by column chromatography on silica gel (1.5 g, 0%–0.5% acetone in petroleum ether) to provide **23** as a colorless oil (110.8 mg, 68%). R_f_ = 0.71 (petroleum ether/acetone 95:5).

IR (KBr) ν 2930, 2865, 1464, 1379, 1369, 1255, 1097, 883, 849, 678 cm^−1^.

^1^H NMR (400 MHz, CDCl_3_): δ 5.47–5.36 (symmetrical m, 2H), 4.27 (tt, 1H, *J* = 6.4, 5.6 Hz), 4.06 (dd, 1H, *J* = 8.2, 6.4 Hz), 3.83 (broad tt, 1H, *J* = 5.9, 5.5 Hz), 3.73 (dd, 1H, *J* = 8.2, 6.4 Hz), 3.54–3.39 (m, 4H with 1H dd at 3.52 ppm, *J* = 9.9, 5.6 Hz and 1H dd at 3.42 ppm, *J* = 9.9, 5.6 Hz), 2.29–2.20 (m, 2H), 2.07–1.96 (m, 2H), 1.62–1.53 (m, 2H plus signal of water as a singlet at 1.59 ppm), 1.53–1.39 (m, 5H including CH_3_, q, 3H at 1.42 ppm, *J* = 0.6 Hz), 1.37 (q, 3H, *J* = 0.6 Hz, CH_3_), 1.36–1.20 (m, 18H), 1.06 (s, 21H), 0.88 (t, 3H, *J* = 6.9 Hz, CH_3_).

^13^C NMR (100 MHz, CDCl_3_): δ 131.40 (CH=CH), 125.66 (CH=CH), 109.37 (CMe_2_), 74.76 (CH), 72.27 (CH), 71.89 (CH_2_), 71.83 (CH_2_), 66.94 (CH_2_), 36.55 (CH_2_), 34.68 (CH_2_), 31.92 (CH_2_), 29.71 (CH_2_), 29.64 (CH_2_), 29.57 (CH_2_), 29.53 (CH_2_), 29.47 (CH_2_), 29.34 (CH_2_), 27.55 (CH_2_), 26.78 (CH_3_), 26.07 (CH_2_), 25.43 (CH_3_), 24.81 (CH_2_), 22.64 (CH_2_), 18.22 (6 CH_3_), 14.11 (CH_3_), 12.64 (3 CH).

[α]_D_^19^: +4.1; [α]_578_^19^: +4.2; [α]_546_^19^: +4.9; [α]_436_^19^: +8.2; [α]_365_^19^: +12.4 (c 3.62, CHCl_3_).

[α]_D_^19^: +4.0; [α]_578_^19^: +3.9; [α]_546_^19^: +4.5; [α]_436_^19^: +7.9; [α]_365_^19^: +12.2 (c 2.51, acetone).

HRMS (ESI, *m*/*z*) calculated for C_33_H_66_O_4_NaSi [M + Na]^+^: 577.4628, found: 577.4627.

Elementary analysis calculated for C_33_H_66_O_4_Si: C, 71.42; H, 11.99, found: C, 71.88; H, 12.09.

#### 3.2.12. (*R*,*Z*)-18-(((*R*)-2,2-Dimethyl-1,3-dioxolan-4-yl)methoxy)octadec-9-en-7-ol (**24**)

To a stirred solution at of **23** (278 mg, 0.5 mmol) in anhydrous THF (1.5 mL) under nitrogen, which was cooled at -20 °C, was added a 1 M solution of TBAF in THF (0.67 mL, 0.67 mmol, 1.3 equiv). The reaction mixture was then left under stirring for 20 h at rt. TLC monitoring showed completion of the reaction, and distilled water (1.5 mL) was added to the mixture. Extraction was done with EtOAc and organic layers were washed with brine and dried over Na_2_SO_4_. Solvent was removed under reduced pressure, and the crude product was purified by column chromatography on silica gel (2.5 g, 0%–4% acetone in petroleum ether) to provide **24** as a colorless oil (184.2 mg, 92%). *R*_f_ = 0.18 (petroleum ether/acetone 95:5).

IR (KBr) ν 3457, 2928, 2856, 1466, 1370, 1256, 1214, 1120, 846, 724 cm^−1^.

^1^H NMR (400 MHz, CDCl_3_): δ 5.56 (dtt, 1H, *J* = 10.9, 7.3, 1.5 Hz), 5.40 (dtt, 1H, *J* = 10.9, 7.3, 1.5 Hz), 4.27 (dddd, 1H, *J* = 6.4, 6.4, 5.7, 5.7 Hz), 4.06 (dd, 1H, *J* = 8.2, 6.4 Hz, ddd with improving the resolution, *J* = 8.2, 6.4, 0.2 Hz), 3.73 (dd, 1H, *J* = 8.2, 6.4 Hz), 3.66–3.56 (m, 1H), 3.54–3.39 (m, 4H with 1H dd at 3.52 ppm, *J* = 9.9, 5.7 Hz (ddd with improving the resolution, *J* = 9.9, 5.7, 0.3 Hz) and 1H dd at 3.42 ppm, *J* = 9.9, 5.6 Hz), 2.24–2.18 (m, 2H), 2.05 (br dtd (apparent qd), 2H, *J* = 7.2, 7.2, 1 Hz), 1.61–1.52 (m, 3H, CH_2_ and OH), 1.51–1.43 (m, 2H), 1.43 (q, 3H, *J* = 0.6 Hz, CH_3_), 1.37 (q, 3H, *J* = 0.6 Hz, CH_3_), 1.37–1.23 (m, 18H), 0.88 (t, 3H, *J* = 6.9 Hz, CH_3_).

^13^C NMR (100 MHz, CDCl_3_): δ 133.49 (CH=CH), 125.16 (CH=CH), 109.37 (CMe_2_), 74.76 (CH), 71.87 (CH_2_), 71.83 (CH_2_), 71.51 (CH), 66.94 (CH_2_), 36.86 (CH_2_), 35.37 (CH_2_), 31.85 (CH_2_), 29.66 (CH_2_), 29.55 (CH_2_), 29.46 (CH_2_), 29.41 (CH_2_), 29.36 (CH_2_), 29.25 (CH_2_), 27.42 (CH_2_), 26.78 (CH_3_), 26.04 (CH_2_), 25.73 (CH_2_), 25.43 (CH_3_), 22.63 (CH_2_), 14.10 (CH_3_).

[α]_D_^20^: -5.0; [α]_578_^20^: −5.4; [α]_546_^20^: −12.5; [α]_436_^20^: −11.9; [α]_365_^20^: −21.2 (c 6.00, CHCl_3_).

Elementary analysis calculated for C_24_H_46_O_4_: C, 72.31; H, 11.63, found: C, 71.79; H, 11.66.

#### 3.2.13. (*S*)-3-(((*R*,*Z*)-12-Hydroxyoctadec-9-en-1-yl)oxy)propane-1,2-diol (**11**)

To a solution of **24** (1.26 g, 3.15 mmol) in methanol (12.6 mL), was added *p*-toluenesulfonic acid monohydrate (30 mg, 0.16 mmol, 0.05 eq) and distilled water (1.26 mL). The flask was then purged with nitrogen, stoppered and dipped in a preheated bath at 60 °C. Stirring was maintained for 4 h at 60 °C. TLC monitoring showed completion of the reaction, and sodium bicarbonate (14.5 mg, 0.173 mmol, 0.055 equiv) was added and stirring was continued for 1 h at 60 °C. Methanol was removed under reduced pressure, and the crude product was purified by column chromatography on silica gel eluting with 0%–5% acetone in petroleum ether to remove non polar impurities and then with petroleum ether + 5% acetone + 12% methanol to give **11** as a colorless oil (945.6 mg, 84%). R_f_ = 0.02 (petroleum ether/acetone 80:20).

IR (KBr) ν 3383, 2926, 2855, 1654, 1466, 1124, 1045, 858, 724 cm^−1^.

^1^H NMR (400 MHz, CDCl_3_): δ 5.56 (dtt, 1H, *J* = 10.8, 7.3, 1.4 Hz), 5.40 (dtt, 1H, *J* = 10.7, 7.4, 1.5 Hz), 3.91–3.81 (m, 1H), 3.72 (very broad dd, 1H, *J* = 10, 4 Hz (br dd after exchange with D_2_O, *J* = 11.3, 3.5 Hz), 1H of CH_2_OH), 3.69–3.58 (m, 2H, 1H of CH_2_OH and H_12_), 3.57–3.41 (m, 4H, CH_2_OCH_2_), 2.80–2.50 (envelope, 1H, CHOH), 2.42–2.12 (envelope, 1H, CH_2_OH), 2.21 (broad ddd, 2H, *J* = 7.4, 6.2, 1.1 Hz), 2.05 (br dt (apparent q), 2H *J* = 7.0, 6.9 Hz), 1.90–1.51 (m, 6H with an envelope centered at 1.63 ppm), 1.51–1.41 (m, 3H), 1.40–1.19 (m, 18H), 0.88 (pseudo t, 3H, *J* = 6.8 Hz, CH_3_).

^13^C NMR (100 MHz, CDCl_3_): δ 133.50 (CH=CH), 125.18 (CH=CH), 72.53 (CH_2_), 71.79 (CH_2_), 71.52 (CH), 70.40 (CH), 64.31 (CH_2_), 36.83 (CH_2_), 35.36 (CH_2_), 31.85 (CH_2_), 29.61 (CH_2_), 29.53 (CH_2_), 29.38 (CH_2_), 29.36 (CH_2_), 29.32 (CH_2_), 29.17 (CH_2_), 27.39 (CH_2_), 26.02 (CH_2_), 25.73 (CH_2_), 22.63 (CH_2_), 14.10 (CH_3_).

[α]_D_^22^: +1.2; [α]_578_^22^: +0.5; [α]_546_^22^: +0.6; [α]_436_^22^: +1.3; [α]_365_^22^: +1.6 (c 1.115, acetone).

HRMS (ESI, m/z) calculated for C_21_H_42_O_4_Na [M + Na]^+^: 381.2981, found: 381.2982.

#### 3.2.14. Methyl (*Z*)-12-oxooctadec-9-enoate (**25**)

Pyridium chlorochromate (18.06 g, 83.8 mmol, 2.6 equiv) was suspended in DCM (111.7 mL) with stirring for 5 min. Following this, a solution of methyl ricinoleate **13** (10 g, 32 mmol) in DCM (15 mL) was added rapidly to the mixture and the flask containing **13** was rinsed with DCM (3 x 2 mL). Stirring was pursued for 1 h at rt under nitrogen. TLC monitoring showed completion of the reaction, and petroleum ether/EtOAc (90:10) (111.7 mL) was added. The resulting mixture was filtered over a short plug of silica gel with rinsing of silica gel by petroleum ether/EtOAc (90:10). Evaporation of the filtrate under reduced pressure followed by the purification of the crude by column chromatography on silica gel (55 g, 0%–1% acetone in petroleum ether) afforded **25** as a colorless oil (6.73 g, 68%). R_f_ = 0.36 (petroleum ether/acetone 90:10).

^1^H NMR (400 MHz, CDCl_3_): δ 5.62–5.50 (m, 2H), 3.67 (s, 3H), 3.15 (br d, 2H, *J* = 6.0 Hz), 2.43 (dd, 2H, *J* = 7.5, 7.4, Hz), 2.30 (dd, 2H, *J* = 7.7, 7.4 Hz), 2.02 (br dt (apparent q), 2H, *J* = 7.5, 7.0 Hz), 1.67–1.51 (m, 4H), 1.40–1.20 (m, 14H), 0.88 (t, 3H, *J* = 6.8 Hz, CH_3_).

^13^C NMR (100 MHz, CDCl_3_): δ 209.36 (CO), 174.31(CO), 133.53 (CH=CH), 120.99 (CH=CH), 51.45 (CH_3_), 42.37 (CH_2_), 41.64 (CH_2_), 34.04 (CH_2_), 31.59 (CH_2_), 29.25 (CH_2_), 29.11 (CH_2_), 29.06 (CH_2_), 29.05 (CH_2_), 28.88 (CH_2_), 27.46 (CH_2_), 24.90 (CH_2_), 23.77 (CH_2_), 22.49 (CH_2_), 14.04 (CH_3_).

#### 3.2.15. Methyl (*Z*)-12,12-difluorooctadec-9-enoate (**26**)

To a solution of **25** (6 g 19.3 mmol) in DCM (22.5 mL) with stirring and under nitrogen at rt, was added dropwise DAST (6.22 mL, 47 mmol, 2.4 equiv). The corresponding mixture was stirred for 21 days at rt. TLC monitoring showed that the reaction was mostly done and saturated aqueous sodium bicarbonate (32 mL) plus water (20 mL) were added to quench unreacted DAST. After partitioning and extraction of the aqueous layer with DCM, combined organic layers were washed with brine and dried over Na_2_SO_4_. DCM was removed under reduced pressure, and the crude product was purified by column chromatography on silica gel (20 g). First, elution with 0%–0.5% acetone in petroleum ether) afforded **26** as a colorless oil (3.48 g, 54%). R_f_ = 0.39 (petroleum ether/acetone 95:5). Then unreacted **25** (2.00 g, 33%) was eluted with 1% acetone in petroleum ether.

IR (KBr) ν 3465, 3021, 2953, 2930, 2856, 1742, 1467, 1436, 1198, 1170, 876, 726 cm^−1^.

^1^H NMR (400 MHz, CDCl_3_): δ 5.64–5.55 (m, 1H), 5.39 (dtt, 1H, *J* = 10.9, 7.3, 1.5 Hz), 3.67 (s, 3H), 2.65–2.52 (m, which could be analyzed as a td (*J_HF_* = 15.9 Hz, *J_HH_* = 7.3 Hz) with further very small couplings, 2H), 2.30 (dd, 2H, *J* = 7.7, 7.4 Hz), 2.03 (br dt (apparent q), 2H, *J* = 7, 7 Hz), 1.87–1.72 (m, 2H), 1.67–1.57 (m, 2H), 1.51–1.41 (m, 2H), 1.40–1.22 (m, 14H), 0.89 (t, 3H, *J* = 6.8 Hz, CH_3_).

^13^C NMR (100 MHz, CDCl_3_): δ 174.32 (CO), 134.52 (CH=CH), 124.87 (C_12_, t, *J* = 241.2 Hz), 120.30 (CH=CH, t, *J* = 5.8 Hz), 51.47 (CH_3_), 35.98 (CH_2_, t, *J* = 25.0 Hz), 34.62 (CH_2_, t, *J* = 26.4 Hz), 34.09 (CH_2_), 31.60 (CH_2_), 29.30 (CH_2_), 29.14 (CH_2_), 29.09 (CH_2_), 29.08 (CH_2_), 29.06 (CH_2_), 27.40 (CH_2_), 24.93 (CH_2_), 22.51 (CH_2_), 22.16 (CH_2_, t, *J* =4.6 Hz), 14.04 (CH_3_).

^19^F NMR (376 MHz, CDCl_3_): δ −96.88 (pentuplet, *J* = 16.3 Hz on ^19^F-undecoupled spectrum).

#### 3.2.16. (*Z*)-12,12-Difluorooctadec-9-en-1-ol (**27**)

To a stirred solution of **26** (199.5 mg, 0.6 mmol) in anhydrous Et_2_O (2 mL) under nitrogen, which was cooled at 0 °C, was added dropwise a solution of Red-Al (65% in toluene, ~3 M, 0.3 mL, 0.9 mmol, 1.5 equiv); then stirring was continued for 5 h at 0 °C. TLC monitoring confirmed completion of the starting material. A solution of citric acid (400 mg) in water (5 mL) was added, and stirring was still continued for 30 min. Extraction was done with petroleum ether/EtOAc (80:20), and organic layers were dried over Na_2_SO_4_. Solvent was evaporated under reduced pressure, and the crude product was purified by column chromatography on silica gel (4 g, 0%–2% acetone in petroleum ether) to afford **27** as a colorless oil (141.3 mg, 77%). R_f_ = 0.21 (petroleum ether/acetone 85:15). This reduction was subsequently performed on a larger scale (15 ×), and the crude alcohol, which was thus obtained, was used as such for the next step.

^1^H NMR (400 MHz, CDCl_3_): δ 5.65–5.56 (m, 1H), 5.39 (dtt, 1H, *J* = 10.9, 7.4, 1.6 Hz), 3.64 (t, 2H, *J* = 6.6 Hz), 2.65–2.53 (m, which could be analyzed as a tdd with further very small couplings at 2.59 ppm, 2H, *J_HF_* = 15.9 Hz, *J_HH_* = 7.3, 1.4 Hz), 2.04 (br dt (apparent q), 2H, *J* = 7, 7 Hz), 1.87–1.72 (m, 2H), 1.61–1.52 (m, 2H), 1.50–1.41 (m, 2H), 1.40–1.23 (m, 16H), 0.89 t, 3H, *J* = 6.9 Hz, CH_3_).

^13^C NMR (100 MHz, CDCl_3_): δ 134.59 (CH=CH), 124.89 (CF_2_, t, *J* = 241.2 Hz), 120.25 (t, CH=CH, *J* = 5.8 Hz), 63.07 (CH_2_), 35.97 (t, CH_2_, *J* = 25.0 Hz), 34.62 (t, CH_2_, *J* = 26.4 Hz), 32.78 (CH_2_), 31.60 (CH_2_), 29.47 (CH_2_), 29.38 (CH_2_), 29.36 (CH_2_), 29.22 (CH_2_), 29.06 (CH_2_), 27.42 (CH_2_), 25.73 (CH_2_), 22.51 (CH_2_), 22.16 (t, CH_2_, *J* = 4.6 Hz), 14.04 (CH_3_).

^19^F NMR (376 MHz, CDCl_3_): δ −96.85 (pentuplet, *J* = 16.3 Hz on ^19^F-undecoupled spectrum).

HRMS (ESI, *m*/*z*) calculated for C_18_H_34_OF_2_Na [M + Na]^+^: 327.2475 found: 327.2478, calculated for C_18_H_33_OFNa [M – HF + Na]^+^: 307.2413, found: 307.2415.

#### 3.2.17. (*Z*)-12,12-Difluorooctadec-9-en-1-yl methanesulfonate (**28**)

To a stirred solution of crude **27** (made from 3.09 g of **26**, 9.3 mmol) and Et_3_N (1.95 mL, 14.0 mmol, 1.5 equiv) in DCM (28 mL) under nitrogen, which was cooled at −35 °C, was added dropwise mesyl chloride (0.9 mL, 11.6 mmol, 1.25 equiv) in DCM (3.7 mL). Transfer of mesyl chloride was completed by rinsing with DCM (3 × 0.2 mL). The reaction mixture was then allowed to warm up slowly to −5 °C (in 4–5 h) and TLC monitoring (elution with DCM) showed completion of the reaction. Distilled water (96 mL) was added to quench the reaction, and extraction of the aqueous layer was done with DCM. Combined organic layers were washed with brine and dried over Na_2_SO_4_. DCM was removed under reduced pressure, and the crude product was purified by column chromatography on silica gel (10 g, 0%–1% acetone in petroleum ether) to afford **28** as a colorless oil (2.44 g, 69% from **26**). R_f_ = 0.61 (petroleum ether/acetone 80:20).

^1^H NMR (400 MHz, CDCl_3_): δ 5.65–5.55 (m, 1H), 5.39 (dtt, 1H, *J* = 10.9, 7.3, 1.6 Hz), 4.22 (t, 2H, *J* = 6.6 Hz), 3.00 (s, 3H), 2.65–2.52 (m, which could be analyzed as a tdd with further very small couplings at 2.59 ppm, 2H, *J_HF_
*= 16.0 Hz, *J_HH_
*= 7.3, 1.3 Hz), 2.04 (br dt (apparent q), 2H, *J* = 7, 7 Hz), 1.87–1.70 (m, 4H), 1.58–1.21 (m, 18H), 0.89 (t, 3H, *J* = 6.9 Hz, CH_3_).

^13^C NMR (100 MHz, CDCl_3_): δ 134.51 (CH=CH), 124.87 (C_12_, t, *J* = 241.2 Hz), 120.30 (t, CH=CH, *J* = 5.8 Hz), 70.16 (CH_2_), 37.36 (CH_3_), 35.97 (t, CH_2_, *J* = 25.0 Hz), 34.61 (t, CH_2_, *J* = 26.4 Hz), 31.60 (CH_2_), 29.31 (CH_2_), 29.29 (CH_2_), 29.13 (CH_2_), 29.12 (CH_2_), 29.05 (CH_2_), 28.98, (CH_2_), 27.39 (CH_2_), 25.40 (CH_2_), 22.50 (CH_2_), 22.16 (t, CH_2_, *J* = 4.6 Hz), 14.05 (CH_3_).

^19^F NMR (376 MHz, CDCl_3_): δ −96.86 (pentuplet, *J* = 16.3 Hz on ^19^F-undecoupled spectrum).

#### 3.2.18. (*R*,*Z*)-4-(((12, 12-Difluorooctadec-9-en-1-yl)oxy)methyl)-2,2-dimethyl-1,3-dioxolane (**29**)

To a stirred mixture of **28** (115 mg, 0.3 mmol), *n*-Bu_4_NBr (24.2 mg, 0.075 mmol, 0.25 equiv), DMSO (0.6 mL) and 50% aqueous NaOH (63 µL, 1.2 mmol of NaOH, 4 equiv), was added (*R*)-solketal **18** (49 mg, ≥95% pure, 0.35 mmol, 1.17 equiv). The corresponding mixture was stirred for 5 h at 60 °C. TLC monitoring showed completion of the reaction, and distilled water was added. Extraction was done with petroleum ether/EtOAc (80:20). Organic layers were washed again with brine and dried over Na_2_SO_4_. Solvent was removed under reduced pressure, and the crude product was purified by column chromatography on silica gel (5 g, 0%–0.5% acetone in petroleum ether) to provide **29** as a colorless oil (79.3 mg, 63%). R_f_ = 0.44 (petroleum ether/acetone 95:5).

^1^H NMR (400 MHz, CDCl_3_): δ 5.64–5.56 (m, 1H), 5.38 (dtt, 1H, *J* = 10.9, 7.3, 1.6 Hz), 4.27 (dddd (apparent tt), 1H, *J* = 6.4, 6.4, 5.7, 5.7 Hz), 4.06 (dd, 1H, *J* = 8.2, 6.4 Hz), 3.73 (dd, 1H, *J* = 8.2, 6.4 Hz), 3.54–3.39 (m, 4H), 2.65–2.53 (m, which could be analyzed as a tdd with further very small couplings at 2.59 ppm, 2H, *J_HF_* = 15.9 Hz, *J_HH_* = 7.4, 1.3 Hz), 2.03 (br dt (apparent q), 2H, *J* = 7, 7 Hz), 1.87–1.72 (m, 2H), 1.62–1.51 (m, 2H), 1.43 (q, 3H, *J* = 0.6 Hz), 1.37 (q, 3H, *J* = 0.6 Hz), 1.40–1.21 (m, 18H), 0.89 (t, 3H, *J* = 6.8 Hz, CH_3_).

^13^C NMR (100 MHz, CDCl_3_): δ 134.59 (CH=CH), 124.88 (CF_2_, t, *J* = 241.2 Hz), 120.23 (t, CH=CH, *J* = 5.8 Hz), 109.37 (CMe_2_), 74.76 (CH), 71.87 (CH_2_), 71.84 (CH_2_), 66.93 (CH_2_), 35.96 (t, CH_2_, *J* = 25.0 Hz), 34.62 (t, CH_2_, *J* = 26.4 Hz), 31.60 (CH_2_), 29.55 (CH_2_), 29.45 (CH_2_), 29.41 (CH_2_), 29.37 (CH_2_), 29.23 (CH_2_), 29.06 (CH_2_), 27.43 (CH_2_), 26.78 (CH_3_), 26.05 (CH_2_), 25.43 (CH_3_), 22.50 (CH_2_), 22.15 (t, CH_2_, *J* = 4.6 Hz), 14.04 (CH_3_).

^19^F NMR (376 MHz, CDCl_3_): δ −96.85 (pentuplet, *J* = 16.3 Hz on ^19^F-undecoupled spectrum).

HRMS (ESI, *m*/*z*) calculated for C_24_H_44_O_3_F_2_Na [M + Na]^+^: 441.3156, found: 441.3149, calculated for C_24_H_43_O_3_FNa [M – HF + Na]^+^: 421.3094, found: 421.3100, calculated for C_24_H_42_O_3_Na [M – 2HF + Na]^+^: 401.3032, found: 401.3044.

#### 3.2.19. (*S*,*Z*)-3-((12,12-Difluorooctadec-9-en-1-yl)oxy)propane-1,2-diol (**9**)

To a solution of acetonide **29** (971.9 mg, 2.32 mmol) in methanol (9.3 mL) and distilled water (0.93 mL) was added *p*-toluenesulfonic acid monohydrate (22.1 mg, 0.116 mmol, 0.05 equiv). The flask was then purged with nitrogen, stoppered and dipped in a preheated bath at 60 °C. Stirring was maintained for 5 h at 60 °C. TLC monitoring showed completion of the reaction, and sodium bicarbonate (10.9 mg, 0.13 mmol, 0.056 equiv) was added to the mixture and the stirring was continued for 1 h at 60 °C. Methanol was removed under reduced pressure, and then the crude product was purified by column chromatography on silica gel (6 g, 0%–5% acetone in petroleum ether and then petroleum ether + 5% acetone + 12% methanol) to provide **9** as a green oil (811.2 mg, 92%). R_f_ = 0.04 (petroleum ether/acetone 90:10).

^1^H NMR (400 MHz, CDCl_3_): δ 5.65–5.55 (m, 1H), 5.39 (dtt, 1H, *J* = 10.9, 7.3, 1.6 Hz), 3.91–3.82 (m (ddt after exchange with D_2_O, *J* = 6.0, 5.2, 3.9 Hz), 1H), 3.72 (ddd, 1H, *J* = 11.4, 6.9, 3.8 Hz (dd after exchange with D_2_O, *J* = 11.4, 3.9 Hz)), 3.65 (ddd, 1H, *J* = 11.4, 5.1, 5.0 Hz (dd after exchange with D_2_O, *J* = 11.4, 5.2 Hz)), 3.56–3.42 (m with 1H dd at 3.54 ppm, *J* = 9.7, 3.9 Hz, 4H), 2.67 (d, which was suppressed after exchange with D_2_O, 1H, *J* = 5.1, Hz, OH), 2.65–2.53 (m, which could be analyzed as a tdd with further very small couplings at 2.59 ppm, 2H, *J_HF_
*= 15.9 Hz, *J_HH_
*= 7.3, 1.2 Hz, 2H), 2.26 (br dd, 1H, *J* = 6.6, 5.6 Hz), 2.04 (br dt (apparent q), 2H, *J* = 7, 7 Hz), 1.87–1.73 (m, 2H), 1.58 (br tt, 2H, *J* = 7.2, 6.7 Hz), 1.51–1.41 (m, 2H), 1.40–1.22 (m, 16H), 0.89 (t, 3H, *J* = 6.9 Hz, CH_3_).

^13^C NMR (100 MHz, CDCl_3_): δ 134.58 (CH=CH), 124.89 (CF_2_, t, *J* = 241.2 Hz), 120.25 (t, CH=CH, *J* = 5.8 Hz), 72.53 (CH_2_), 71.82 (CH_2_), 70.41 (CH), 64.30 (CH_2_), 35.96 (t, CH_2_, *J*= 25.0 Hz), 34.62 (t, CH_2_, *J* = 26.4 Hz), 31.60 (CH_2_), 29.57 (CH_2_), 29.44 (CH_2_), 29.40 (CH_2_), 29.36 (CH_2_), 29.22 (CH_2_), 29.05 (CH_2_), 27.42 (CH_2_), 26.07 (CH_2_), 22.50 (CH_2_), 22.15 (t, CH_2_, *J*= 4.6 Hz), 14.05 (CH_3_).

^19^F NMR (376 MHz, CDCl_3_): δ −96.84 (pentuplet, *J*= 16.3 Hz on ^19^F-undecoupled spectrum).

[α]_D_^24^: −4.6; [α]_578_^24^: −5.8; [α]_546_^24^: −6.4; [α]_436_^24^: −9.9; [α]_365_^24^: −14.3 (c 0.94, acetone).

HRMS (ESI, *m*/*z*) calculated for C_21_H_40_O_3_F_2_Na [M + Na]^+^: 401.2843, found: 401.2840, calculated for C_21_H_39_O_3_FNa [M – HF + Na]^+^: 381.2781, found: 381.2791, calculated for C_21_H_38_O_3_Na [M – 2HF + Na]^+^: 361.2719, found: 361.2726.

#### 3.2.20. Methyl (*R*,*Z*)-12-((methylsulfonyl)oxy)octadec-9-enoate (**31**)

To a stirred solution of **13** (10.0 g, 32.0 mmol) and Et_3_N (9.15 mL, 65.5 mmol, 2.05 equiv) in DCM (80 mL), which was cooled under nitrogen cooled at −40 °C, mesyl chloride (5.0 mL, 64.0 mmol, 2.0 equiv) in DCM (10 mL) was added dropwise. Transfer of mesyl chloride was completed by rinsing with DCM (2 × 1 mL). The reaction mixture was allowed to warm up slowly to −10 °C and then further stirred for 4 h at −10 °C. TLC monitoring showed completion of the reaction, and distilled water (120 mL) was added. Extraction of the aqueous layer was done with DCM. Combined organic layers were washed with brine and dried over Na_2_SO_4_. DCM was removed under reduced pressure, and the crude product was purified by column chromatography on silica gel (36 g, 0%–1% acetone in petroleum ether) to afford **31** as a colorless oil (8.06 g, 65%). R_f_ = 0.26 (petroleum ether/acetone 80:20).

^1^H NMR (400 MHz, CDCl_3_): δ 5.55 (dtt, 1H, *J* = 10.9, 7.3, 1.5 Hz), 5.37 (dtt, 1H, *J* = 10.9, 7.2, 1.5 Hz), 4.69 (tt, 1H, *J* = 6.2, 6.1 Hz), 3.67 (s, 3H), 2.99 (s, 3H), 2.55–2.37 (m, 2H), 2.30 (t, 2H, *J* = 7.5 Hz), 2.03 (br dt (apparent q), 2H, *J* = 7, 6.5 Hz), 1.72–1.57 (m, 4H), 1.49–1.18 (m, 16H), 0.88 (t, 3H, *J* = 6.9 Hz, CH_3_).

^13^C NMR (100 MHz, CDCl_3_): δ 174.32 (C_quat_, CO), 133.81 (CH=CH), 123.03 (CH=CH), 83.66 (CH), 51.47 (CH_3_), 38.68 (CH_3_), 34.25 (CH_2_), 34.08 (CH_2_), 32.54 (CH_2_), 31.64 (CH_2_), 29.39 (CH_2_), 29.14 (CH_2_), 29.10 (CH_2_), 29.08 (CH_2_), 29.00 (CH_2_), 27.42 (CH_2_), 25.05 (CH_2_), 24.92 (CH_2_), 22.56 (CH_2_), 14.05 (CH_3_).

[α]_D_^18^: +12.3; [α]_578_^18^: +12.5; [α]_546_^18^: +14.2; [α]_436_^18^: +24.3; [α]_365_^18^: +38.6 (c 4.00, acetone).

[α]_D_^18^: +10.6; [α]_578_^18^: +11.0; [α]_546_^18^: +12.6; [α]_436_^18^: +21.5; [α]_365_^18^: +33.7 (c 4.00, CHCl_3_).

#### 3.2.21. Methyl (*S*,*Z*)-12-azidooctadec-9-enoate (**32**)

To a solution of **31** (9.1 g, 23.3 mmol) in DMSO (34.9 mL), was added NaN_3_ (2.6 g, 40.0 mmol, 1.7 equiv). The flask was then purged with nitrogen, stoppered and dipped in a preheated bath at 80 °C. Stirring was maintained for 16 h at the same temperature. TLC monitoring showed completion of the reaction, and it was brought back to rt, then quenched with an aqueous solution of NH_4_Cl. Extraction was done with petroleum ether/EtOAc (80:20, 185 mL), and organic layers were washed with brine and dried over Na_2_SO_4_. Solvent was removed under reduced pressure to obtain the crude product as a light-yellow oil. The crude product was then purified by column chromatography on silica gel (30 g, 0%–0.5% acetone in petroleum ether) to provide **32** as an oil (6.26 g, 80%). R_f_ = 0.58 (petroleum ether/acetone 80:20).

IR (KBr) ν 3011, 2929, 2855, 2100, 1742, 1456, 1250, 1197, 726 cm^−1^.

^1^H NMR (400 MHz, CDCl_3_): δ 5.52 (dtt, 1H, *J* = 10.9, 7.3, 1.5 Hz), 5.38 (dtt, 1H, *J* = 10.9, 7.2, 1.5 Hz), 3.67 (s, 3H), 3.33–3.25 (m, which could be analyzed as a dddd at 3.29 ppm with *J*= 7.6, 6.5, 6.5, 5.3 Hz, 1H, CHN_3_), 2.33–2.25 (m, 4H with a dd at 2.34 ppm integrating for 2 protons, *J* = 7.7, 7.4 Hz), 2.04 (br td (apparent q), 2H, *J* = 7.0, 6.8 Hz), 1.68–1.57 (m, 2H), 1.56–1.40 (m, 3H), 1.40–1.22 (m, 15H), 0.89 (t, 3H, *J* = 6.9 Hz, CH_3_).

^13^C NMR (100 MHz, CDCl_3_): δ 174.33 (C_quat_, CO), 133.10 (CH=CH), 124.58 (CH=CH), 62.95 (CH), 51.46 (CH_3_), 34.10 (CH_2_), 33.99 (CH_2_), 32.28 (CH_2_), 31.72 (CH_2_), 29.46 (CH_2_), 29.15 (CH_2_), 29.10 (2 CH_2_), 29.09 (CH_2_), 27.40 (CH_2_), 26.13 (CH_2_), 24.93 (CH_2_), 22.59 (CH_2_), 14.07 (CH_3_).

#### 3.2.22. (*S*,*Z*)-12-Azidooctadec-9-enal (**33**)

To a vigorously stirred solution of **32** (5.063 g, 15 mmol) in Et_2_O (60 mL), which was cooled at −80 °C under nitrogen, was added dropwise a solution of Dibal (25% in hexane, 28 mL, 34.5 mmol, 2.3 equiv). The resulting mixture was allowed to stir for 2 h at −80 °C. TLC monitoring showed completion of the reaction. A saturated aqueous solution of potassium sodium tartrate (25 mL) was added, and stirring was continued for 10 min. Extraction was then done with petroleum ether/EtOAc (80:20). Organic layers were washed with brine and dried over Na_2_SO_4_. Solvent was evaporated under reduced pressure, and the crude product was used as such for the next step. For analytical purposes, the crude reaction product, which was initially obtained from 10 times less product, was purified by column chromatography on silica gel (5.2 g, 0%–1% acetone in petroleum ether) to afford **33** as an ochre oil (313.1 mg from 506.3 mg of **32**, 68%). R_f_ = 0.4 (petroleum ether/acetone 85:15).

IR (KBr) ν 34,340, 2929, 2856, 2716, 2100, 1727, 1466, 1273, 725 cm^−1^.

^1^H NMR (400 MHz, CDCl_3_): δ 9.77 (t, 1H, *J* = 1.9 Hz, CHO), 5.52 (dtt, 1H, *J* = 10.9, 7.2, 1.5 Hz, CH=CH–CH_2_–CHN_3_), 5.39 (dtt, 1H, *J* = 10.9, 7.2, 1.5 Hz, CH=CH–CH_2_–CHN_3_), 3.33–3.25 (m, which could be analyzed as a dddd at 3.29 ppm with *J* = 7.6, 6.6, 6.5, 5.2 Hz, 1H, CHN_3_), 2.42 (td, 2H, *J* = 7.4, 1.9 Hz, CH_2_CHO), 2.35–2.20 (m, 2H, CHCH_2_CH=CH), 2.05 (br td (apparent q), 2H, *J* = 7.0, 6.8 Hz, CH=CHCH_2_), 1.68–1.58 (m, 2H, CH_2_CH_2_CHO), 1.57–1.47 (m, 2H, CH_2_CHN_3_), 1.47–1.41 (m, 1H, of CH_2_CH_2_CHN_3_), 1.41–1.24 (m, 15H, 1H of CH_2_CH_2_CHN_3_, CH_3_(CH_2_)_3_ and (CH_2_)_4_CH_2_CH_2_CHO), 0.89 (t, 3H, *J* = 6.9 Hz, CH_3_).

^13^C NMR (100 MHz, CDCl_3_): δ 202.92 (CHO), 133.05 (CH=CH), 124.62 (CH=CH), 62.94 (CH–N_3_), 43.91 (CH_2_), 33.99 (CH_2_), 32.29 (CH_2_), 31.72 (CH_2_CH_2_CH_3_), 29.44 (CH_2_), 29.25 (CH_2_), 29.11 (CH_2_), 29.08 (CH_2_), 29.07 (CH_2_), 27.39 (CH_2_), 26.13 (CH_2_), 22.58 (CH_2_CH_3_), 22.05 (CH_2_), 14.07 (CH_3_).

[α]_D_^18.5^: −31.5; [α]_578_^18.5^: −33.4; [α]_546_^18.5^: −38.2; [α]_436_^18.5^: −67.5; [α]_365_^18.5^: −111.0 (c 2.00, acetone).

[α]_D_^18.5^: −24.8; [α]_578_^18.5^: −25.9; [α]_546_^18.5^: −29.7; [α]_436_^18.5^: −52.0; [α]_365_^18.5^: −85.4 (c 2.60, CHCl_3_).

Elementary analysis calculated for C_18_H_33_N_3_O: C, 70.31; H, 10.82; N, 13.67 found: C, 70.39; H, 11.01; N, 13.38.

#### 3.2.23. (*S*,*Z*)-12-Azidooctadec-9-en-1-ol (**34**)

To a vigorous stirred solution of **33** (crude made from 5.063 g, 15 mmol of **32**) in ethanol (15 mL), which was cooled at 0 °C under nitrogen, was added NaBH_4_ (435 mg, 11.25 mmol, 0.75 equiv). The resulting mixture was allowed to stir for 1 h at 0 °C. TLC monitoring showed completion of the reaction. About 10–20 drops of acetone were added, and ethanol was removed under reduced pressure. After addition of water, extraction with petroleum ether/EtOAc (80:20), organic layers were washed with brine and dried over Na_2_SO_4_. Solvent was removed under reduced pressure, and the crude product was purified by two successive column chromatographies on silica gel (15 g, 0%–2% acetone in petroleum ether) to afford **34** as an ochre oil (3.44 g, 74% from **32**). *R*_f_ = 0.36 (petroleum ether/acetone 85:15).

^1^H NMR (400 MHz, CDCl_3_): δ 5.52 (dtt, 1H, *J* = 10.9, 7.3, 1.5 Hz), 5.38 (dtt, 1H, *J* = 10.9, 7.3, 1.5 Hz), 3.64 (t, 2H, *J* = 6.6 Hz), 3.33–3.25 (m, which could be analyzed as a dddd at 3.29 ppm with *J* = 7.8, 6.8, 6.4, 5.2 Hz, 1H, CHN_3_), 2.35–2.20 (m, 2H), 2.04 (br td (apparent q), 2H, *J* = 7.0, 7.0 Hz), 1.61–1.41 (m, 5H), 1.41–1.23 (m, 17H), 0.89 (t, 3H, *J* = 6.9 Hz, CH_3_).

^13^C NMR (100 MHz, CDCl_3_): δ 133.17 (CH=CH), 124.53 (CH=CH), 63.07 (CH_2_), 62.96 (CH), 33.98 (CH_2_), 32.79 (CH_2_), 32.28 (CH_2_), 31.72 (CH_2_), 29.51 (CH_2_), 29.49 (CH_2_), 29.39 (CH_2_), 29.24 (CH_2_), 29.09 (CH_2_), 27.43 (CH_2_), 26.13 (CH_2_) 25.73 (CH_2_), 22.59 (CH_2_), 14.07 (CH_3_).

#### 3.2.24. (S,Z)-12-Azidooctadec-9-en-1-yl methanesulfonate (**35**)

To a stirred solution of **34** (3.93 g, 12.7 mmol) and Et_3_N (2.7 mL, 19.1 mmol, 1.5 equiv) in DCM (38 mL), which was cooled at –50 °C under nitrogen, was added dropwise mesyl chloride (1.25 mL, 15.9 mmol, 1.25 equiv) in DCM (5.0 mL). Transfer of mesyl chloride was completed by rinsing with DCM (3 × 0.2 mL). The mixture was then allowed to stir for 4 h at −50 °C and then allowed to warm up slowly to −5 °C. TLC monitoring (elution with DCM) showed completion of the reaction, and distilled water (50 mL) was added to quench the reaction. Extraction was done with DCM, and organic layers were washed with brine and dried over Na_2_SO_4_. DCM was removed under reduced pressure, and the crude product was purified by two successive column chromatographies on silica gel (15 g and then 40 g, 0%–3% acetone in petroleum ether) to provide **35** as an ochre oil (2.89 g, 59%). R_f_ = 0.61 (petroleum ether/acetone 80:20).

^1^H NMR (400 MHz, CDCl_3_): δ 5.52 (dtt, 1H, *J* = 10.9, 7.3, 1.4 Hz), 5.39 (dtt, 1H, *J* = 10.9, 7.3, 1.5 Hz), 4.22 (t, 2H, *J* = 6.6 Hz), 3.33–3.25 (m, which could be analyzed as a br dddd at 3.29 ppm with *J* = 7.2, 6.8, 6.5, 5.3 Hz, 1H, CHN_3_), 3.01 (s, 3H), 2.35–2.22 (m, 2H), 2.05 (br dt (apparent q), 2H, *J* = 7.2, 7.0 Hz), 1.79–1.70 (m, 2H), 1.56–1.22 (m, 20H with 2H at 1.56–1.47 ppm), 0.89 (t, 3H, *J* = 6.9 Hz, CH_3_).

^13^C NMR (100 MHz, CDCl_3_): δ 133.07 (CH=CH), 124.60 (CH=CH), 70.17 (CH_2_), 62.94 (CH), 37.37 (CH_3_), 33.99 (CH_2_), 32.29 (CH_2_), 31.72 (CH_2_), 29.47 (CH_2_), 29.31 (CH_2_), 29.16 (CH_2_), 29.12 (CH_2_), 29.08 (CH_2_), 28.99 (CH_2_), 27.40 (CH_2_), 26.13 (CH_2_), 25.41 (CH_2_), 22.58 (CH_2_), 14.07 (CH_3_).

#### 3.2.25. (*R*)-4-((((*S*,*Z*)-12-Azidooctadec-9-en-1-yl)oxy)methyl)-2,2-dimethyl-1,3-dioxolane (**36**)

To a stirred mixture at rt of **35** (2.48 g, 6.4 mmol), *n*-Bu_4_NBr (515.8 mg, 1.6 mmol, 0.25 equiv), DMSO (12.8 mL) and 50% aqueous NaOH (0.67 mL, 2 equiv) was added (*R*)-solketal **18** (1002.8 mg, ≥95% pure, 7.21 mmol, 1.13 equiv). The resulting mixture was then stirred overnight for 15 h at 35 °C. TLC monitoring showed completion of the reaction, and distilled water (40 mL) was added. Extraction was done with petroleum ether/EtOAc (80:20), and organic layers were washed again with brine and dried over Na_2_SO_4_. Solvent was removed under reduced pressure, and the crude product was purified by two successive column chromatographies on silica gel (11 g and then 30 g, 0%–0.5% acetone in petroleum ether) to afford **36** as an ochre oil (1.41 g, 52%). R_f_ = 0.56 (petroleum ether/acetone 95:5).

^1^H NMR (400 MHz, CDCl_3_): δ 5.52 (dtt, 1H, *J* = 10.9, 7.2, 1.5 Hz), 5.38 (dtt, 1H, *J* = 10.9, 7.2, 1.5 Hz), 4.27 (tt, 1H, *J* = 6.4, 5.7 Hz), 4.06 (dd, 1H, *J* = 8.2, 6.4 Hz), 3.73 (dd, 1H, *J* = 8.2, 6.4 Hz), 3.54–3.39 (m, 4H with 1H dd at 3.52 ppm, *J* = 9.9, 5.7 Hz and 1H dd at 3.42 ppm, *J* = 9.9, 5.6 Hz), 3.33–3.25 (m, 1H, CHN_3_), 2.36–2.20 (m, 2H), 2.04 (br td (apparent q), 2H, *J* = 7.1, 7.0 Hz), 1.62–1.40 (m, 7H, 2 CH_2_ + CH_3_ q at 1.43 ppm, *J* = 0.6 Hz), 1.40–1.24 (m, 21H, 9 CH_2_ + CH_3_ q at 1.37 ppm, *J* = 0.6 Hz), 0.89 (t, 3H, *J* = 6.9 Hz, CH_3_).

^13^C NMR (100 MHz, CDCl_3_): δ 133.18 (CH=CH), 124.51 (CH=CH), 109.37 (CMe_2_), 74.76 (CH), 71.88 (CH_2_), 71.83 (CH_2_), 66.94 (CH_2_), 62.95 (CH), 33.98 (CH_2_), 32.28 (CH_2_), 31.72 (CH_2_), 29.55 (CH_2_), 29.52 (CH_2_), 29.48 (CH_2_), 29.42 (CH_2_), 29.26 (CH_2_), 29.08 (CH_2_), 27.44 (CH_2_), 26.78 (CH_3_), 26,13 (CH_2_), 26.05 (CH_2_), 25.43 (CH_3_), 22.58 (CH_2_), 14.07 (CH_3_).

HRMS (ESI, *m*/*z*) calculated for C_24_H_45_N_3_O_3_Na [M + Na]^+^: 446.3358, found: 446.3365.

#### 3.2.26. (*S*)-3-(((*S*,*Z*)-12-Azidooctadec-9-en-1-yl)oxy)propane-1,2-diol (**10**)

To a solution of **36** (610.7 mg, 1.44 mmol) in methanol (5.8 mL), was added *p*-toluenesulfonic acid monohydrate (13.7 mg, 0.072 mmol, 0.05 equiv) and distilled water (0.58 mL). The flask was then purged with nitrogen, stoppered and dipped in a preheated bath at 60 °C. Stirring was maintained for 4 h at 60 °C. TLC monitoring showed completion of the reaction, and sodium bicarbonate (6.7 mg, 0.08 mmol, 0.055 equiv) was added to the mixture, and stirring was continued for 1 h at 60 °C. Methanol was removed under reduced pressure, and the crude product was purified by column chromatography on silica gel (5 g, 0%–5% acetone in petroleum ether and then petroleum ether + 5% acetone + 12% methanol) to provide **10** as an ochre oil (539.1 mg, 97%). R_f_ = 0.03 (petroleum ether/acetone 80:20).

IR (KBr) ν 3396, 2928, 2855, 2100, 1464, 1254, 1125, 1037 cm^−1^.

^1^H NMR (400 MHz, CDCl_3_): δ 5.52 (dtt, 1H, *J* = 10.9, 7.3, 1.4 Hz), 5.38 (dtt, 1H, *J* = 10.9, 7.2, 1.5 Hz), 3.91–3.83 (m, which could be analyzed as a br tt at 3.86 ppm with *J* = 5.3, 4.1 Hz, exchange with D_2_O improved a little bit the resolution), 3.77–3.68 (m, 1H (dd at 3.71 ppm after exchange with D_2_O, *J* = 11.4, 3.9 Hz)), 3.65 (dd, 1H, *J* = 11.4, 5.2 Hz, unchanged after exchange with D_2_O), 3.56–3.42 (m, 4H with 1H dd at 3.54 ppm, *J* = 9.7, 4.0 Hz), 3.33–3.25 (m, which could be analyzed as a dddd at 3.29 ppm with *J* = 7.6, 6.6, 6.4, 5.2 Hz, 1H, CHN_3_), 2.72 (envelope from 2.85 to 2.59 ppm, 1H, OH), 2.37–2.20 (m, 3H: 1 OH as a partly overlapped envelope, which topped at 2.32 ppm + 1 CH_2_, which was centered at 2.29 ppm), 2.05 (br dt (apparent q), 2H, *J* = 7.1, 6.9 Hz), 1.62–1.40 (m, 5H), 1.40–1.23 (m, 17H), 0.89 (t, 3H, *J* = 6.9 Hz, CH_3_).

^13^C NMR (100 MHz, CDCl_3_): δ 133.16 (CH=CH), 124.53 (CH=CH), 72.52 (CH_2_), 71.83 (CH_2_), 70.43 (CH), 64.29 (CH_2_), 62.95 (CH), 33.97 (CH_2_), 32.28 (CH_2_), 31.72 (CH_2_), 29.57 (CH_2_), 29.51 (CH_2_), 29.46 (CH_2_), 29.41 (CH_2_), 29.24 (CH_2_), 29.08 (CH_2_), 27.43 (CH_2_), 26.13 (CH_2_), 26.07 (CH_2_), 22.58 (CH_2_), 14.07 (CH_3_).

[α]_D_^21^: −17.7; [α]_578_^21^: −18.7; [α]_546_^21^: −21.3; [α]_436_^21^: −37.9; [α]_365_^21^: −62.5 (c 2.26, CHCl_3_).

HRMS (ESI, *m*/*z*) calculated for C_21_H_41_N_3_O_3_Na [M + Na]^+^: 406.3046, found: 406.3046, calculated for C_21_H_40_N_3_O_3_Na_2_ [M − H + 2Na]^+^: 428.2865, found: 428.2882, calculated for C_21_H_41_NO_3_Na [M – N_2_ + Na]^+^: 378.2984, found: 378.2998.

### 3.3. Chemicals and Culture Media

Nystatin (Maneesh Pharmaceutic PVT) for fungi, chloramphenicol (Sigma) for *E. coli* ATCC 10.536 and AG102, gentamycin (Jinling Pharmaceutic Group corp.), tetracycline, ciprofloxacin and ampicillin (Sigma-Aldrich, South Africa) for bacteria were used as reference antibiotics (RE). Dimethylsulfoxide (DMSO, Sigma) was used to dilute all tested samples. Nutrient agar (NA) was used for bacterial culture. Sabouraud glucose agar was used for the activation of the fungi. The Mueller Hinton broth (MHB) was used to determine the minimal inhibition concentration (MIC) of all samples against the tested microorganisms. Two-fold dilutions were made for MIC determinations, and the results were validated only if at least two of the three replications were similar; standard deviation bars were not appropriate in regards to two-fold dilutions in such study.

### 3.4. Microbial Strains

The organisms tested included methicillin-resistant *Staphylococcus aureus* LMP805, *Streptococcus faecalis* LMP 806 (Gram-positive bacteria), Gram-negative bacteria, namely *β*-lactamase positive (βL^+^) *Escherichia coli* LMP701*, E. coli* ATCC10536*,* kanamycin and chloramphenicol resistant *E. coli AG102*, ampicillin-resistant *Shigella dysenteriae*LMP820, chloramphenicol-resistant *Salmonella typhi* LMP706, chloramphenicol-resistant *Citrobacter freundii* LMP802 and three fungi, namely *Candida albicans* LMP709U, *Candida glabrata* LMP0413U and *Microsporum audouinii* LMP725D. *E. coli* ATCC10536 and AG102 were provided by UMR-MD1 (Université de la Méditeranée, France). All other microbial species were clinical isolates from the “Centre Pasteur du Cameroun” and provided by the Laboratory of Applied Microbiology and Molecular Pharmacology (LMP) (Faculty of Science, University of Yaoundé I). These were maintained on agar slant at 4 °C and sub-cultured on a fresh appropriate agar plate 24 h prior to any antimicrobial test. The three types of *E. coli* used in the antimicrobial assay were the reference ATCC strain, a wild type and a resistant phenotype.

### 3.5. Antimicrobial Assays

The antimicrobial assays were conducted using rapid XTT colorimetry and viable count methods. The XTT colorimetric assay was performed according to Pettit et al. [36] as modified by Kuete et al. [37,38,39]. Concisely, the tested sample (or combined sample with antibiotic) was first of all dissolved in DMSO/MHB. The final concentration of DMSO was lower than 1% and did not affect the microbial growth [37,38,39]. The solution obtained was then added to MHB and serially diluted two fold (in a 96-well microplate). Then, 100 µl of inoculum 1.5 × 10^6^ CFU/mL was prepared in MHB. The plates were covered with a sterile plate sealer, then agitated to mix the contents of the wells using a plate shaker and incubated at 30 °C for 48 h (*M. audouinii*) or at 37 °C for 24 h (other organisms). The assay was repeated three times. Gentamicin, chloramphenicol (bacteria) and nystatin (fungi) were used as positive controls. Wells containing MHB, 100 µl of inoculum and DMSO to a final concentration of 1% served as negative controls. The MIC of samples was then detected following an addition (40 µL) of 0.2 mg/mL *p*-iodonitrotetrazolium chloride and incubated at 37 °C for 30 min. Viable bacteria reduced the yellow dye to a pink color. MIC was defined as the lowest sample concentration that prevented this change and exhibited complete inhibition of bacterial growth. 

Bacterial enumeration was performed on *E. coli* LMP701 as described by Stenger et al. [40]. Cells were treated with samples at their MIC and 4× MIC values as previously determined using XTT assay, and incubated at 37 °C. Viable cells were then determined at 0, 30, 60, 120, 240 and 480 min by performing 10-fold serial dilutions of this suspension in 0.9% saline. Gentamicin was used as reference drug whilst 0.9% saline and DMSO to a final concentration of 1% was used as control. All dilutions were placed on nutrient agar plates that were then incubated at 37 °C for 18 h. Bacterial colonies were then enumerated, and the total CFU/mL at each time was deduced.

## 4. Conclusions

A series of novel 1-*O*-alkylglycerol compounds **8**–**11** were synthesized from cheap ricinoleic acid (**12**). The structures of these compounds were characterized by NMR experiments as well as from the HRMS and elementary analysis data. AKGs **8**–**11** were evaluated for their respective antimicrobial activities. All compounds exhibited antimicrobial activity to different extents alone. Additionally, some beneficial synergistic effects were observed when AKG **8** was combined with gentamicin, and when AKG **11** was combined with ciprofloxacin and ampicillin. AKG **11** was viewed as a lead compound for this series, as it exhibited significant antimicrobial activity alone and when combined with some antibiotics compared with **8**–**10**. It is now evident that non-natural synthesized 1-*O*-alkylglycerols can be further explored as a new source of drugs, and can be used in diverse preparations of pharmaceutical importance.

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
