# Peer review of "Synthesis and Biological Evaluation of Four New Ricinoleic Acid-Derived 1-O-alkylglycerols"

_marinedrugs, 2020, doi:10.3390/md18020113_

Round 1
Reviewer 1 Report
Reviewer 1: comment 7:Why does the authors did not tried Red-Al for the conversion of 32-34? How does azide group interacts with red-al to not for the desired product? Any clarification?
Author Response
The response was upload as a PDF file

Reviewer 2 Report
The re-submitted manuscript was adequately improved from the original manuscript to publish on Marine Drugs. Therefore, the reviewer wants to support the publication of this paper.
If possible, please add the elemental analysis of compound 28.
In addition, it seems to be better to add the structure of compound18 in Scheme 1 of the revised version.
Author Response
Response was upload as a PDF file

Reviewer 3 Report
Changes Accepted!
Author Response
Response was Added as a PDF file

Reviewer 4 Report
As the authors have complied with most of my (and other reviewers') comments, I recommend their paper for the publication in Marine Drugs.
Author Response
Response was upload as a PDF file

This manuscript is a resubmission of an earlier submission. The following is a list of the peer review reports and author responses from that submission.
Round 1
Reviewer 1 Report
The manuscript entitled " Synthesis and antimicrobial evaluation of novel methoxyl. difluorinated, azide, and hydroxyl substituted 1-O-alkylglycerols from ricinoleic acid derivative" by Pemha R et al. describes the synthesis of ricenoleic acid and their evaluation towards antimicrobials. Please see comments below:
1) The title of the manuscript needs to be concised.
2) Fig. 2 is often confuse the readers, please make two structures and specify double bond positions and define as 2-5 and 6,7 as separately.
3) The authors have introduce the idea of derivatives in line 77, this needs to be moved to line 88 -93.
4) In scheme 4, Does the authors treated compound 13 with Red-Al to access product 21, please give rationale.
5) In scheme 6, the reaction conditions of C is missing (conversion of to 26 to 27) and also subsequent steps, the letter d is repeated, please correct the scheme 6.
6) In scheme 6, 28 to 29, Does the authors have tried 1-4 hour monitoring to see only the formation of acetonide 29? Please repeat and include the results.
7) Does the authors tried Redl-Al to convert 32 to 34, if so, please provide rationale.
8) In vitro studies were performed with the derivatives of ricenoleic acid as synthesized with other known microbials. How about treating all the microbials with ricinoleic acid? Please provide the results to see the effect of o-alkylglycerols. Discuss them.
9) lines 478 and 498 has a typo, please correct them. Please check throughout the manuscript for errors.
10) please provide HRMS for 28.
11) Please correct the title for figure S 4.13 as proton NMR (page 57 of Supplementary material)
please address the issues as major revisions
Reviewer 2 Report
The manuscript described by Pemha and Mosset reports the synthesis of four new O-alkylglycerols derived from ricinoleic acid and the assay using these compounds for antimicrobial activity. Although the introduction part is well-written, the number of the compounds are insufficient and the SAR study seems to be immature to publish.
The authors should use not only (S)-glycerol derivative but also (R)-glycerol derivative. As compound 11 has the highest activity, the epimer of 11 should be tested. The epimer of 11 can be synthesized from compound 13 via Mitsunobu reaction. Taking the results in table 1 into account, the hydrogen bonding donor/acceptor seems to be important for the antimicrobial activity. Therefore, the authors recommended to reduce azido group of 10 to amine, which could act as hydrogen bond donor/acceptor.
The standard errors could not be found in tables 1 and 2. How many times were the experiments investigated? How about reproducibility? In addition, the authors were recommended the control experiments using ricinoleic acid, its acyl glycerol derivative and/or desilylated 21.
In Scheme 3, 4 conditions were described for the reduction of 15. Since there is no rational discussion on this reduction, the reviewer considers that the non-optimized reductions can be essentially deleted. In addition, the order of reduction conditions in scheme 3 and the order in experimental part (page 12) are inconsistent. In Scheme 8, the authors found unique half-reduction of ester using DIBAL-H in Et2O. Did the authors check the reproducibility? The authors used various bases for SN2 reaction with compound 18. Please explain the reason in the main text.
Overall, the reviewer considered that this manuscript did not meet the criteria of Marine Drugs.
Reviewer 3 Report
Line 230 to 258. Could you please replace the number 9,10,11 etc with the name of the sample? It would be easy to follow the discussions.
Why the authors used 3 types of E.coli? Which is the explanation of different MIC for Ec1,2,3?Could you please insert in the discussion section an explanation?
For figures 4 and 5 could you add also the standard deviation bars?
Line 863. please delete double dots CFU./.mL
Reviewer 4 Report
In their paper, the authors report the synthesis and biological evaluation of four new ricinoleic acid-derived 1-O-alkylglycerols. I think the presented findings are interesting and deserve for publication in Marine Drugs, however, the manuscript should be corrected.
1) First of all, authors should try to make their manuscript more concise. In my opinion, the 'Introduction' section is too long. Moreover, in the 'Results and Discussion' section, the reader can find some introductory paragraphs. For example, in lines 98-111; 138-143; 167-170, authors try to motivate their choice of target molecules by citing relevant literature. Ok, this should be done in the 'Introduction' section and should be concise.
2) Full systematic nomenclature of target molecules in the Abstract should be avoided (lines 29-32). I think the 'Experimental section' is a good place to show such names.
3) Reaction schemes should be cleaned up. Size of molecular structures should be the same in each scheme. Every single synthetic route should be presented in one scheme. For example scheme 1 and 3 should be merged (similarly: schemes 4 and 5; schemes 6 and 7; schemes 8-11 should also be merged).
4) Scheme 3 is missing.
5) Reagent listings in all schemes should be carefully checked because there are many editorial mistakes. For example in Scheme 1: there is BF3.MeOH but should be BF3·MeOH (line 121); in Scheme 3: there is NaBH4/LiCl but should be NaBH4/LiCl. And so on.
6) There are also several unclear things in the 'Experimental section'. HRMS for compound 16 is apparently incorrect (line 401) (16 has 19 carbon atoms but in HRMS molecular formula we have 38 carbon atoms). In line 547, authors report pseudomolecular ion [M2+Na+] with 35 carbon atoms. So, the question arises - how many carbon atoms does M itself have?
7) For compound 34, one carbon atom is missing in 13C NMR peak listing.
8) What is the purpose of reporting several values of the specific rotation for the single compound?
9) Copies of all 1H and 13C NMR (together with 19F NMR for compound 9) spectra should be attached to the manuscript in a separate file.
10) In the reduction of an ester 15 to primary alcohol 16 poor yields were reported when LiAlH4 was used as reducing agent. In the 'Experimental section', the authors state that starting material was completely consumed, so why the yield is so low? Were there any difficulties in isolation of the product from reaction mixture?